# Bidirectional Seismic Energy Input to an Isotropic Nonlinear One-Mass Two-Degree-of-Freedom System

Kenji Fujii

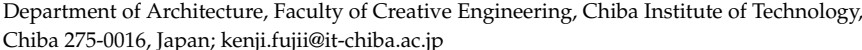

Department of Architecture, Faculty of Creative Engineering, Chiba Institute of Technology, Chiba 275-0016, Japan; kenji.fujii@it-chiba.ac.jp

**Abstract:** The test results obtained for reinforced concrete columns by several studies have revealed that the peak displacement and cumulative hysteresis energy are important parameters for evaluating the damage of columns under horizontal bidirectional and unidirectional loading. Therefore, the seismic parameters related to the nonlinear peak displacement and cumulative hysteresis energy with regard to horizontal bidirectional seismic input should be investigated. In this study, the bidirectional seismic input to an isotropic nonlinear one-mass two-degree-of-freedom system was evaluated. First, a dimensionless parameter $\gamma$, which controls the low-cycle fatigue effect, was formulated as a function of two energy input parameters (the maximum momentary input energy and total input energy) and a nonlinear system (ductility and normalized hysteresis energy absorption during a half cycle). Then, the maximum momentary input energy and total input energy were evaluated according to the ground motion characteristics (Fourier coefficient of horizontal ground motion components) and system properties. Finally, the nonlinear peak displacement and parameter $\gamma$ of the nonlinear system were evaluated on the basis of the maximum momentary input energy and total input energy. The results revealed that the nonlinear peak displacement and parameter $\gamma$ can be properly evaluated using two energy parameters.

**Keywords:** bidirectional seismic energy input; peak displacement; cumulative hysteresis energy; complex Fourier series; momentary input energy; total input energy

## 1. Introduction

### 1.1. Background

The peak displacement is an important parameter for the seismic design of building structures. Additionally, it is widely recognized that cumulative damage, or the low-cycle fatigue effect, is also important for the evaluation of seismic damage. During the seismic event, the structural members are subjected to repeated cyclic loading. The structural members under repeated cycle loading may deteriorate further than under monotonic loading due to the low-cyclic fatigue effect. The cumulative hysteresis energy is an important and widely used parameter to represent the low-cycle fatigue effect. To assess the seismic damage of members, the Park–Ang model [1,2] has been widely applied as a damage model. In the Park–Ang model, the structural damage index is defined as a linear combination of the damage caused by the peak displacement and cumulative hysteresis energy. The linear combination assumption has been investigated and verified by Chai at al. [3], who proposed the modification of the original Park–Ang model to avoid problems related to monotonic loading. Another damage model based on the peak displacement and cumulative hysteresis model has been proposed by Poljanšek and Fajfar [4]. Although the Poljanšek–Fajfar model is a nonlinear combination of the damage caused by the peak displacement resulting from the cumulative hysteresis energy, the Poljanšek–Fajfar model is as simple as the Park–Ang model. However, as far as the author's understanding, the Park–Ang model is much more applied than the Poljanšek–Fajfar model because the Park–Ang model has been calibrated by several tests (e.g., [1,3]) and also has a simpler formulation.

To assess the possible damage of a building structure using a damage model, for either the original or modified Park–Ang model, or Poljanšek–Fajfar model, the peak displacement and cumulative energy must be evaluated. The *total input energy* [5,6] is the seismic intensity parameter related to the cumulative damage. Several studies have investigated the seismic demand (peak displacement, total input energy cumulative hysteresis energy) and/or the seismic damage assessment of a building structure based on the damage model [7–43]. In particular, Fajfar [11] proposed the equivalent ductility factor, which considers the effect of cumulative hysteresis energy. This study introduced a dimensionless parameter $\gamma$, which controls the low-cycle fatigue effect. His team also reported that the damage index of the multi-story frame models can be evaluated by simplified nonlinear analysis (*N2 method*) using the $\gamma$ parameter [15]. Additionally, the influence of the duration of ground motion to the total input and cumulative hysteresis energy has been pointed out by several studies [1,9,13,42,43]: in general, the total input energy increases as the duration becomes longer. Several researchers have also assessed the damage of building structures under sequential seismic events applying the Park–Ang model [21,22,24,31,33,35]. Although most of these studies considered unidirectional seismic input [8–38], few studies have considered bidirectional horizontal seismic input [39–43]. Since there are several reports about the horizontal directionality of the ground acceleration (e.g., the rupture directivity effect [44], and the directional dependence of the site effect observed near a basin edge [45]), how to consider such bidirectional horizontal seismic input to damage assessment of a structure is also important.

The contribution of bidirectional (or biaxial) loading to the damage of columns is also a very important issue. Thus far, several test results pertaining to the biaxial horizontal loading of reinforced concrete (RC) columns have been reported [46–52]. Rodrigues et al. made a review of the biaxial loading behavior of RC columns [52]. Qiu et al. [48] and Rodrigues et al. [51] evaluated the damage of RC columns under biaxial horizontally loading, and applied the Park–Ang model [1] to damage evaluation. Their results indicate that the Park–Ang model may be applicable to the damage assessment of RC columns under biaxial horizontal loading.

According to the nonlinear peak displacement, Inoue et al. proposed the *maximum momentary input energy* [53–55] as an intensity parameter related to the peak displacement. The prediction procedure of the nonlinear peak displacement of RC structures subjected to strong unidirectional ground motion has been proposed and verified by analytical [53] and experimental research [54,55]. In these studies, the peak displacement is predicted by equating the maximum momentary input energy and cumulative hysteresis energy during a half cycle of structural response. One important aspect of momentary input energy is that it can be correlated with the total input energy by considering the duration of ground motion. Therefore, it is useful to evaluate the seismic response from the aspect of energy input. From this viewpoint, the authors investigated the relationship between the maximum momentary input energy and the total input energy of an elastic single-degree-of-freedom (SDOF) model [56,57]. Additionally, the concept of the momentary input energy was extended to the bidirectional horizontal excitation considered in previous work [58].

Evaluation of the peak displacement and cumulative hysteresis energy is essential for damage evaluation of a structure subjected to bidirectional horizontal excitation, although there are still many open questions regarding the quantification of RC column damage under biaxial horizontal loading, as noted by Rodrigues et al. [51]. Additionally, the author formulated the time-varying function of the energy input using the Fourier series [57] and extended it to consider an isotropic two-degree-of-freedom model subjected to bidirectional excitation [58]. This formulation indicates that the two seismic intensity parameters, namely the maximum momentary input energy and total input energy, can be evaluated based on the properties of the system and the complex Fourier coefficient of the two horizontal ground motion components.

*1.2. Objectives of This Study*

Based on the above discussion, the following questions are addressed in this paper.

- Can the bidirectional energy input to a nonlinear structure be evaluated from the linear elastic spectrum, as in the case of unidirectional excitation?
- Is the maximum momentary input energy applicable to the evaluation of the bidirectional peak displacement of a nonlinear structure?
- The parameter $\gamma$ is related to the number of cyclic loads. What is the relationship between $\gamma$ and the maximum momentary and total input energy?

In this study, the bidirectional seismic input to an isotropic nonlinear one-mass two-degree-of-freedom system was evaluated. First, the maximum momentary input energy and the total input energy, which are related to the nonlinear peak displacement and cumulative hysteresis energy, respectively, were evaluated based on the ground motion characteristics (Fourier coefficient of horizontal ground motion components) and properties of the two-degree-of-freedom system. Then, the nonlinear peak displacement and a dimensionless parameter, $\gamma$, were predicted using the maximum momentary input energy and cumulative energy input.

The rest of this paper is organized as follows. Section 2 presents the formulation of $\gamma$ using the maximum momentary input energy and total input energy. Then, the evaluation procedure of the nonlinear peak displacement and $\gamma$ is conducted using the maximum momentary input energy and total input energy. The analysis model and ground motion data are presented in Section 3. The validation of the evaluation of (i) the equivalent velocity of the total input energy, (ii) the equivalent velocity of the maximum momentary input energy, (iii) the peak displacement, and (iv) the dimensionless parameter $\gamma$ is discussed in Section 4. Discussions focused on (i) the validation of the model of the increment of the hysteretic dissipated energy in a half cycle, and (ii) the validation of the formulation of $\gamma$ are presented in Section 5.

## 2. Definition and Calculation of Bidirectional Seismic Input Energy

*2.1. Definiton of Bidirectional Momentary and Cumulative Seismic Input Energy for an Isotropic Nonlinear One-Mass Two-Degree-of-Freedom System*

Figure 1 shows the one-mass two-degree-of-freedom model. In this model, $m$ is the mass of the system; $a_{gX}$ and $a_{gY}$ are the X- and Y- components of the horizontal ground acceleration, respectively. The system is assumed to be isotropic, that is the yield surface of the system is assumed to be circular as shown in Figure 1b. In this study, to represent the behavior of ductile RC structures, the envelope of the restoring force–displacement relationship was assumed to be a trilinear curve as shown in Figure 1c. The nonlinear behavior of the system was modelled using the Multi-Shear-Spring (MSS) model proposed by Wada et al. [59].

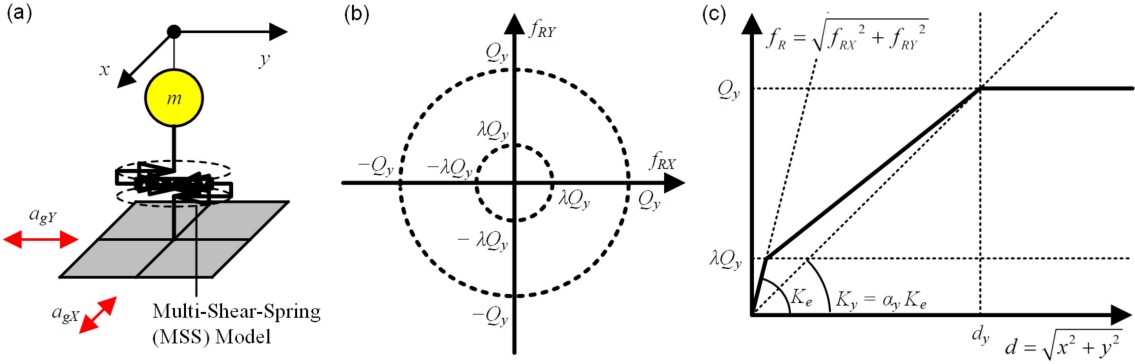

**Figure 1.** An isotropic nonlinear one-mass two-degree-of-freedom model: (**a**) the one-mass two-degree-of-freedom model; (**b**) interaction of yield strength in the X- and Y-directions; (**c**) envelope of the restoring force–displacement relationship.

Let Equation (1) be the system's equation of motion.

$$\mathbf{M}\ddot{\mathbf{d}}(t) + \mathbf{C}(t)\dot{\mathbf{d}}(t) + \mathbf{f_R}(t) = -\mathbf{M}\mathbf{a_g}(t), \tag{1}$$

$$\mathbf{M} = m\begin{bmatrix} 1 & 0 \\ 0 & 1 \end{bmatrix}, \mathbf{f_R}(t) = \left\{\begin{array}{c} f_{RX}(t) \\ f_{RY}(t) \end{array}\right\}, \mathbf{d}(t) = \left\{\begin{array}{c} x(t) \\ y(t) \end{array}\right\}, \mathbf{a_g}(t) = \left\{\begin{array}{c} a_{gX}(t) \\ a_{gY}(t) \end{array}\right\}, \tag{2}$$

where $\mathbf{M}$ is the mass matrix; $\mathbf{C}(t)$ is the damping matrix and is assumed to be proportional to the tangent stiffness matrix; $\mathbf{f_R}(t)$ and $\mathbf{d}(t)$ are the restoring force and displacement vector, respectively; and $\mathbf{a_g}(t)$ is the ground acceleration vector. By multiplying both sides of Equation (1) by $\dot{\mathbf{d}}(t)^{\mathbf{T}}dt$ from the left and integrating from 0 to $t$, the equation of the energy balance from zero to $t$ can be obtained:

$$E_V(t) + E_D(t) + E_S(t) = E_I(t), \tag{3}$$

$$E_V(t) = \int_0^t \dot{\mathbf{d}}(t)^{\mathbf{T}}\mathbf{M}\ddot{\mathbf{d}}(t)dt, E_D(t) = \int_0^t \dot{\mathbf{d}}(t)^{\mathbf{T}}\mathbf{C}(t)\dot{\mathbf{d}}(t)dt, \tag{4}$$

$$E_S(t) = \int_0^t \dot{\mathbf{d}}(t)^{\mathbf{T}}\mathbf{f_R}(t)dt, E_I(t) = -\int_0^t \dot{\mathbf{d}}(t)^{\mathbf{T}}\mathbf{M}\mathbf{a_g}(t)dt, \tag{5}$$

where $E_V(t)$ is the kinetic energy, $E_D(t)$ is the dissipated damping energy, $E_S(t)$ is the strain energy (including the elastic strain energy and hysteresis energy), and $E_I(t)$ is the input energy to the system.

The momentary input energy for the bidirectional excitation, $\Delta E_{BI}$, is defined as follows. According to Inoue et al. [53–55], we consider the energy balance during a half cycle of the structural response (from $t$ to $t + \Delta t$). In this study, the beginning and end time of a half cycle, $t$ and $t + \Delta t$, respectively, are defined as the time when the absolute (vector) value of the displacement $d(t)$ is at a local maximum. The absolute value of displacement $d(t)$ is expressed as:

$$d(t) = |\mathbf{d}(t)| = \sqrt{\{x(t)\}^2 + \{y(t)\}^2}. \tag{6}$$

The conditions for $d(t)$ at a local maximum are expressed as:

$$\left\{\begin{array}{l} \dot{d}(t) = 0 : x(t)\dot{x}(t) + y(t)\dot{y}(t) = 0 \\ \ddot{d}(t) < 0 : x(t)\ddot{x}(t) + y(t)\ddot{y}(t) + \{\dot{x}(t)\}^2 + \{\dot{y}(t)\}^2 < 0 \end{array}\right. . \tag{7}$$

Notably, the definition of the beginning and end time of a half cycle in this study is the same as that in a previous study [58]: for the isotropic linear elastic one-mass two-degree-of-freedom system, the potential energy is its local maximum when the absolute displacement $d(t)$ is its local maximum.

The momentary input energy for the bidirectional excitation, $\Delta E_{BI}$, is defined as:

$$\Delta E_{BI}(t) = -\int_t^{t+\Delta t} \dot{\mathbf{d}}(t)^{\mathbf{T}}\mathbf{M}\mathbf{a_g}(t)dt = -m\int_t^{t+\Delta t} \{a_{gX}(t)\dot{x}(t) + a_{gY}(t)\dot{y}(t)\}dt. \tag{8}$$

The maximum momentary input energy for the bidirectional excitation, $\Delta E_{BI,\max}$, is defined as the maximum value of $\Delta E_{BI}$ over the course of the seismic event. Figure 2 shows the maximum momentary input energy. This figure shows the nonlinear response of the one-mass two-degree-of-freedom system (initial natural period $T_0 = 0.5$ s, ductility $\mu = 2$, input ground motion, Kobe Japan Meteorological Agency Observatory (JMA Kobe)1995 ($\Delta\phi_0 = 0$)). The maximum momentary input energy $\Delta E_{BI,\max}$ is the input energy from $t = 7.86$ s (beginning of half cycle shown in (a)) to $t + \Delta t = 8.40$ s (end of half cycle shown in (a)). By comparing the orbit of the response displacement and restoring force, as shown

in (a) and (b), it is obvious that the shapes of the two orbits are different because of the nonlinearity of the system. Figure 3 shows an example of the hysteresis loop and time-history of the momentary input energy for the case shown in Figure 2. As shown in Figure 3a, the hysteresis loop in the X-direction is similar to that of the unidirectional excitation. However, in the Y-direction, the hysteresis loop is affected by the interaction of the bidirectional strength, as shown in Figure 3b.

The total input energy $E_I$ and cumulative hysteresis energy $E_H$ are defined as the cumulative energy at the end of a seismic event ($t = t_d$), as follows:

$$E_I = E_I(t_d), E_H = E_S(t_d). \tag{9}$$

For convenience, the equivalent velocity of the total energy and the maximum momentary input energy, $V_I$ and $V_{\Delta E}$, are, respectively, defined as:

$$V_I = \sqrt{2E_I/m}, V_{\Delta E} = \sqrt{2\Delta E_{BI,\max}/m}. \tag{10}$$

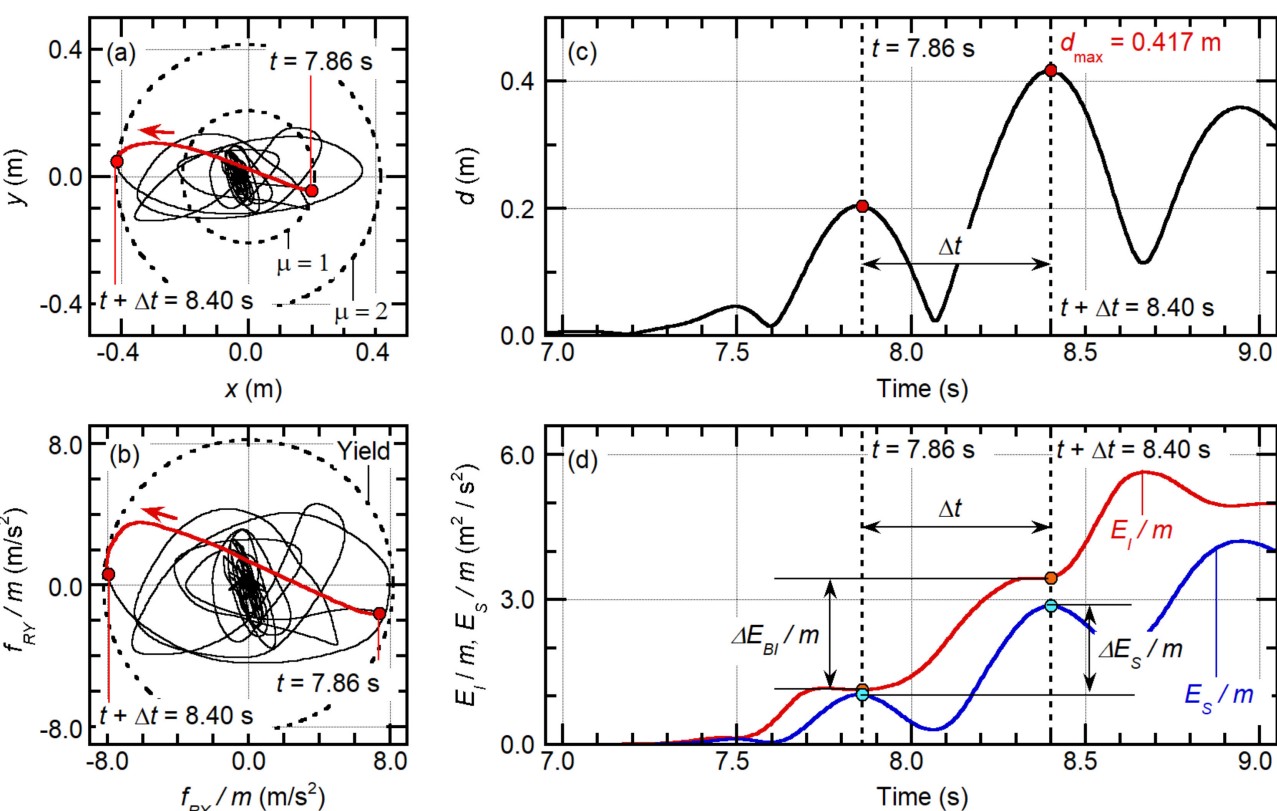

**Figure 2.** Definition of maximum momentary input energy for bidirectional excitation: (**a**) orbit of response displacement; (**b**) orbit of response restoring force; (**c**) time-history of absolute displacement; (**d**) time-history of input energy per unit mass and cumulative strain energy per unit mass.

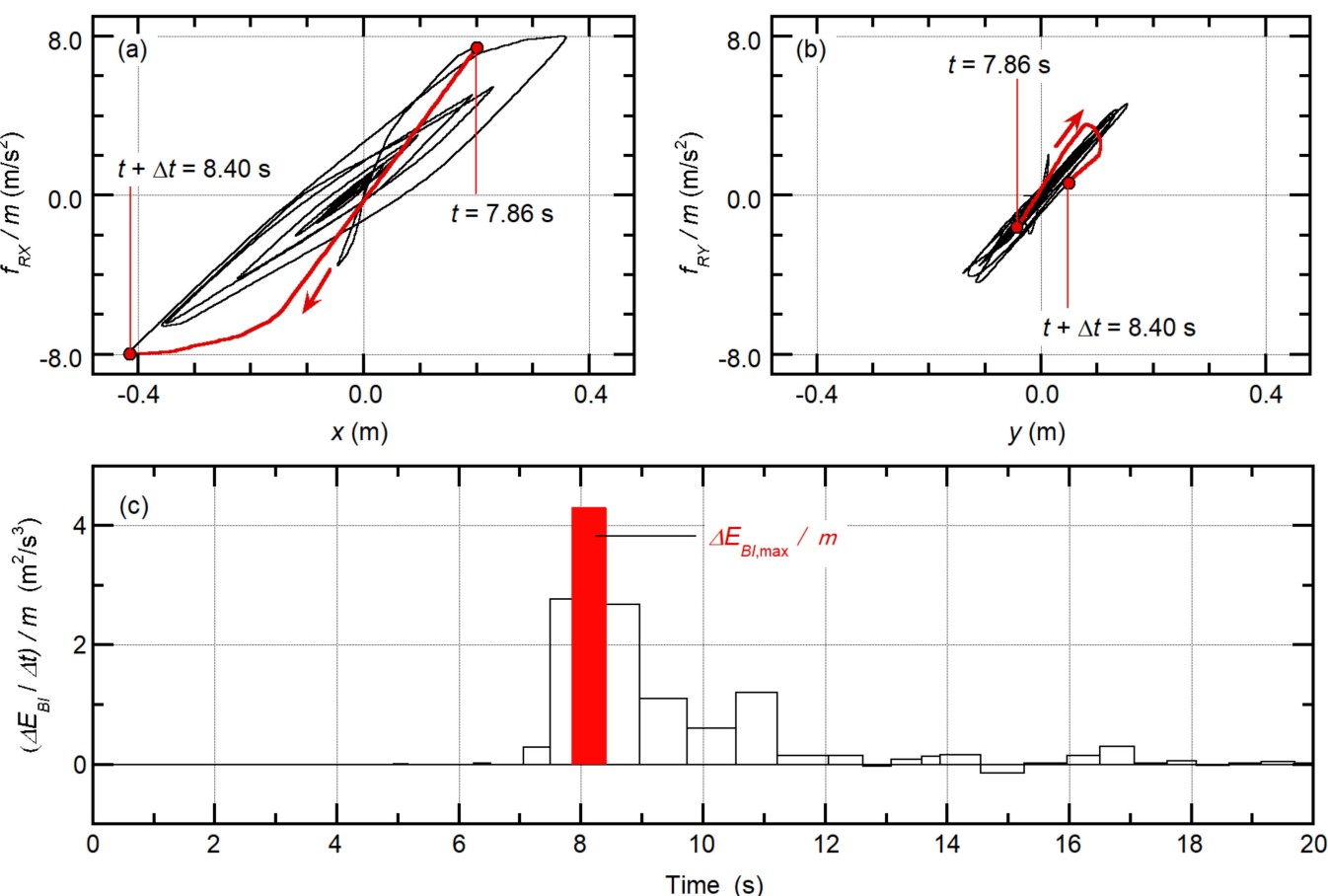

**Figure 3.** Example of hysteresis loop and time-history of momentary input energy: (**a**) the restoring force–displacement relationship in the X-direction; (**b**) the restoring force–displacement relationship in the X-direction; (**c**) time-history of momentary input energy per unit mass.

### 2.2. Formulation of Fajfar's Parameter γ Using Maximum Monentary Input Energy and Cumulative Input Energy

Considering the unidirectional loading of RC columns, the Park–Ang damage index [1] can be written as follows:

$$D = \frac{d_{\max}}{d_{u,M}} + \beta \frac{E_H}{Q_y d_{u,M}}, \tag{11}$$

where $d_{\max}$ and $d_{u,M}$ are the peak displacement and ultimate displacement under monotonic loading, respectively; $Q_y$ and $d_y$ are the yield strength and yield displacement, respectively; and $\beta$ is the positive parameter representing the contribution of hysteresis energy to the structural damage. The first term of Equation (11) represents the damage caused by the large deformation, while the second term represents the damage caused by cyclic loading (low-cycle fatigue). Rodrigues et al. [51] have demonstrated that the damage of RC columns under biaxial loading can be evaluated by using Equation (11), and considering $d_{\max}$ as the absolute value of the peak displacement and $E_H$ as the sum of the cumulative energy of the two principal (loading) axes. Accordingly, the damage of RC structures under bidirectional excitation can be evaluated if the absolute value of peak displacement $d_{\max}$ and cumulative hysteresis energy $E_H$ are properly evaluated.

The dimensionless parameter $\gamma$, which was originally introduced by Fajfar [11], is defined as:

$$\gamma = \frac{1}{\mu} \sqrt{\frac{E_H}{Q_y d_y}}, \tag{12}$$

where $\mu = d_{max}/d_y$ is the ductility of the system. In this study, the yield strength $Q_y$ and yield displacement $d_y$ were assumed to be independent of the angle of horizontal loading. Thus, the definition of $\gamma$ in Equation (12) can be applied to the bidirectional excitation without any modification. According to Fajfar [11], Equation (11) can be rewritten as follows:

$$D = \frac{1}{\mu_{u,M}}\left(\mu + \beta\gamma^2\mu^2\right), \tag{13}$$

where $\mu_{u,M} = d_{u,M}/d_y$ is the ultimate ductility of the system under monotonic loading. Therefore, if the response quantities $\mu$ and $\gamma$ are known, the Park–Ang damage index can be calculated using Equation (13).

The parameter $\gamma$ is related to the number of cyclic loads, while the ratio of the total input energy to the maximum momentary input energy, $E_I/\Delta E_{BI,max}$, represents the value of cyclic energy input. The relationship between $\gamma$ and ratio $V_I/V_{\Delta E}$ is formulated below. According to previous research [53–55], the dissipated hysteresis energy during the half cycle, $\Delta E_\mu$, can be expressed as a function of ductility, as follows:

$$\Delta E_\mu = Q_y d_y f(\mu). \tag{14}$$

The ratio $E_H/\Delta E_\mu$ can be expressed as:

$$\frac{E_H}{\Delta E_\mu} = \frac{(\gamma\mu)^2}{f(\mu)}. \tag{15}$$

Therefore, the parameter $\gamma$ can be expressed as:

$$\gamma = \frac{\sqrt{f(\mu)}}{\mu}\sqrt{\frac{E_H}{\Delta E_\mu}}. \tag{16}$$

Assuming that the peak displacement occurs at the half cycle, when the maximum momentary energy input occurs, $\Delta E_\mu$ is related to $\Delta E_{BI,max}$. For the case in which the damping of the system is proportional to the tangent stiffness (the initial damping ratio $h_0 = 0.05$), Hori and Inoue [55] reported that the ratio of the cumulative hysteresis energy to the maximum momentary input energy of the unidirectional excitation is approximately $(0.85)^2 = 0.7225$. Additionally, according to Fajfar et al. [10], the upper bound of ratio $E_H/E_I$ is approximately 0.8 for a ductile RC structure when $h_0 = 0.05$. In this study, the following assumption was made to simplify the formulation:

$$E_H/\Delta E_\mu \approx E_I/\Delta E_{BI,max} = (V_I/V_{\Delta E})^2. \tag{17}$$

Therefore, the relationship between the parameter $\gamma$ and $V_I/V_{\Delta E}$ is approximated as:

$$\gamma \approx \frac{\sqrt{f(\mu)}}{\mu}\frac{V_I}{V_{\Delta E}}. \tag{18}$$

Equation (18) implies that parameter $\gamma$ depends on the (i) ductility $\mu$ (peak displacement), (ii) normalized dissipated hysteresis energy during the half cycle $f(\mu)$ (fatness of hysteresis loop), (iii) equivalent velocity of total input energy $V_I$, and (iv) equivalent velocity of maximum momentary input energy $V_{\Delta E}$. As ratio $V_I/V_{\Delta E}$ indicates the number of cyclic loads, it is expected that $\gamma$ will be large (the damage caused by the cyclic loading is significant) when $V_I/V_{\Delta E}$ is large.

Notably, the validity of Equation (18) depends on the ductility of the system. As has been shown by Fajfar et al. [10,13], ratio $E_H/E_I$ depends on the ductility. Hence, $E_H/E_I$ increases and approaches a certain value when the ductility is large (in [10], the upper bound is 0.8, as mentioned above). Therefore, two ductility levels ($\mu = 2, 4$) were considered in the numerical investigation, as will be discussed later.

*2.3. Evaluation Procedure for Nonlinear Peak Displacement and Fajfar's Parameter γ Using Maximum Momentary Input Energy and Cumulative Input Energy*

　　In this study, the total input energy $E_I$ and maximum momentary input energy $\Delta E_{BI,\max}$ were evaluated based on the time-varying function of the energy input, which was formulated by the author [58]. It is assumed that the average value of the two ground acceleration components is zero. The formulation of the time-varying function of the momentary energy input is summarized in Appendix A.

　　For the nonlinear isotropic two-degree-of-freedom system (ductility:$\mu \geq 1$), the effective period $T_{eff}$, is defined as:

$$T_{eff} \; = \; \frac{T_y}{3}\left(\frac{1}{\mu} + 2\sqrt{\mu}\right), \tag{19}$$

$$\text{where } T_y \; = \; 2\pi\sqrt{md_y/Q_y} \; = \; T_0/\sqrt{\alpha_y}. \tag{20}$$

　　In this study, the damping was assumed to be proportional to the tangent stiffness; that is, the damping ratio of the system at the initial stage, $h_0$, was assumed to be 0.05. The evaluation procedure is described below.

2.3.1. STEP 1: Calculation of Fourier Coefficient of Ground Motion Components

　　The complex Fourier coefficient of the two horizontal components, $c_{X,n}$ and $c_{Y,n}$, can be, respectively, calculated by the discrete Fourier transform of the two horizontal components $a_{gX}(t)$ and $a_{gY}(t)$.

2.3.2. STEP 2: Calculation of Properties of Equivalent Linear System

　　Let us consider the equivalent linear isotropic two-degree-of-freedom system (effective natural circular frequency $\omega_{eff} = 2\pi/T_{eff}$, equivalent damping $h_{eff}$). From the given properties of the original nonlinear system (mass $m$, initial natural period $T_0$, secant stiffness degradation ratio at yield point $\alpha_y$, ductility $\mu$), the effective period $T_{eff}$ can be calculated using Equation (19). Then, the displacement and velocity transfer function of the equivalent linear system can be calculated as follows:

$$H_D(i\omega_n) \; = \; \frac{1}{\omega_{eff}{}^2 - \omega_n{}^2 + 2h_{eff}\omega_n\omega_{eff}i}, H_V(i\omega_n) \; = \; i\omega_n H_D(i\omega_n). \tag{21}$$

　　In this study, $h_{eff}$ was assumed to be 0.10 for simplicity. This assumption is consistent with the studies by Akiyama [5] and Inoue et al. [53–55].

2.3.3. STEP 3: Calculation of the Time-Varying Function of Momentary Energy Input

　　The duration of the half cycle of response $\Delta t$ can be calculated from the results of STEPS 1 and 2, as follows:

$$\Delta t \; = \; \pi\sqrt{\sum_{n=1}^{N}|H_D(i\omega_n)|^2\left\{|c_{X,n}|^2 + |c_{Y,n}|^2\right\} \Big/ \sum_{n=1}^{N}|H_V(i\omega_n)|^2\left\{|c_{X,n}|^2 + |c_{Y,n}|^2\right\}}. \tag{22}$$

　　Then, the Fourier coefficient of the time-varying function of the momentary energy input can be calculated as follows:

$$E_{\Delta BI,n}{}^* = \begin{cases} \dfrac{\sin(\omega_n \Delta t/2)}{\omega_n \Delta t/2} \displaystyle\sum_{n_1 = n+1}^{N} \{H_V(i\omega_{n_1}) + H_V(-i\omega_{n_1-n})\}\{c_{X,n_1}c_{X,-(n_1-n)} + c_{Y,n_1}c_{Y,-(n_1-n)}\} \\ \qquad : n > 0 \\ 2 \displaystyle\sum_{n_1 = 1}^{N} \mathrm{Re}\{H_V(i\omega_{n_1})\}\{|c_{X,n_1}|^2 + |c_{Y,n_1}|^2\} \\ \qquad : n = 0 \\ \overline{E_{\Delta BI,n}{}^*} \\ \qquad : n < 0 \end{cases} . \tag{23}$$

The momentary input energy per unit mass at time $t$ can be calculated as follows:

$$\frac{\Delta E_{BI}(t)}{m} = \int_{t-\Delta t/2}^{t+\Delta t/2} \sum_{n = -N+1}^{N-1} E_{\Delta BI,n}{}^* \exp(i\omega_n t) dt. \tag{24}$$

The maximum momentary input energy per unit mass, $\Delta E_{BI,\max}/m$, can be evaluated as the maximum value calculated by Equation (24) over the course of the seismic event. The total input energy per unit mass, $E_I/m$, can be calculated as follows:

$$\frac{E_I}{m} = \int_0^{t_d} \sum_{n = -N+1}^{N-1} E_{\Delta BI,n}{}^* \exp(i\omega_n t) dt = t_d E_{\Delta BI,0}{}^*. \tag{25}$$

The equivalent velocity of the maximum momentary input energy and total input energy, $V_I$ and $V_{\Delta E}$, respectively, can be calculated using Equation (10).

### 2.3.4. STEP 4: Calculation of Nonlinear Peak Displacement and Fajfar's Parameter $\gamma$

According to previous research [56], the peak nonlinear displacement $d_{\max}$ can be calculated as follows:

$$d_{\max} = \frac{\mu}{\sqrt{2f(\mu)}} \frac{3}{1/\mu + 2\sqrt{\mu}} \frac{T_{eff}}{2\pi} \left\{\phi(h_0) V_{\Delta E}\left(T_{eff}\right)\right\}, \tag{26}$$

where $\phi(h_0)$ is the ratio of the hysteretic dissipated energy spectrum [53]. In this study, $\phi(h_0)$ was considered to be 0.85, because $h_0$ was assumed to be 0.05.

Figure 4 shows the modelling of the increment in the hysteresis dissipated energy in a half cycle of the ductile RC members. In this study, the model proposed by Nakamura et al. [53] was applied.

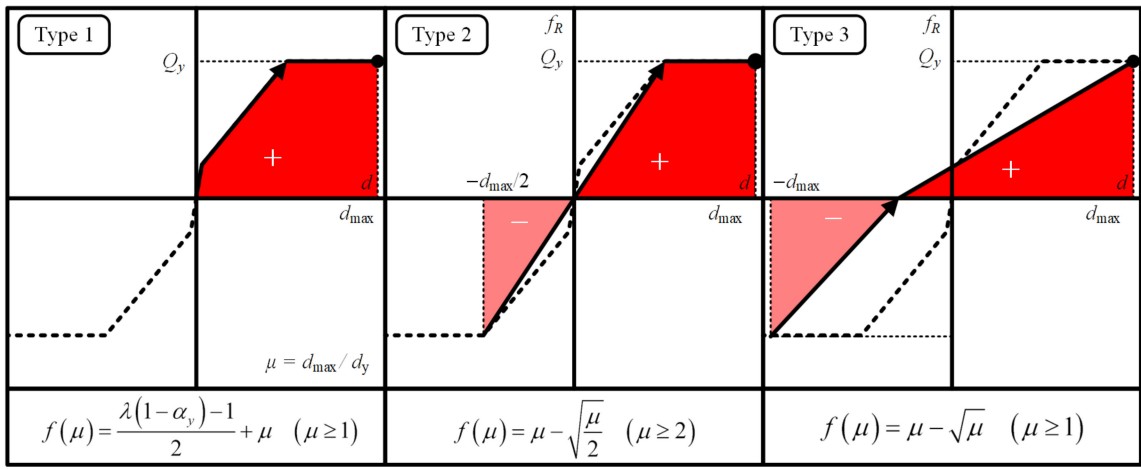

**Figure 4.** Modelling of increment in hysteretic dissipated energy in a half cycle.

The dimensionless parameter $\gamma$ can be calculated from Equation (18). Notably, both $d_{\max}$ and $\gamma$ depend on the assumption of the normalized dissipated hysteresis energy during the half cycle $f(\mu)$. Equations (18) and (26) indicate that $d_{\max}$ increases while $\gamma$ decreases, when the $f(\mu)$ value decreases.

### 3. Analysis Model and Ground Motion Data

#### 3.1. Analysis Model

The analysis model considered in this study is an isotropic one-mass two degree-of freedom system representing ductile RC structures (Figure 1). The envelope curve of the restoring force–displacement relationship is shown in Figure 5a. The hysteresis rule used in the previous study [56] (Figure 5b) was applied to model the nonlinear behavior of ductile RC structures. The Muto model [60] was used with one modification. Specifically, the unloading stiffness after yielding decreases proportionally to $\mu^{-0.5}$, to represent the degradation of unloading.

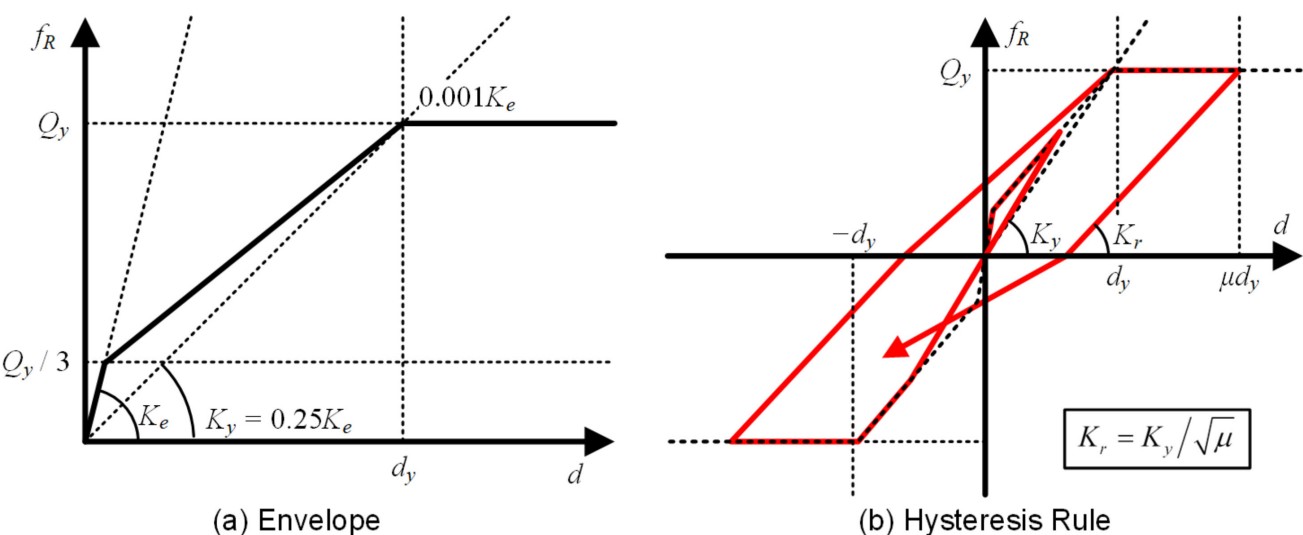

**Figure 5.** Restoring force–displacement relationship of the nonlinear system: (**a**) envelope; (**b**) hysteresis rule.

In the following discussion, the initial period of the system, $T_0$, ranges from 0.10 s to 2.24 s at intervals of 0.02 s when the target ductility is 2.0, and from 0.10 to 1.76 s when the target ductility is 4.0. The reason for this is that the effective period $T_{eff}$ is less than 5.0 s. The yield strength per unit mass $Q_y/m$ of each system is calculated such that the difference between the ductility obtained from the nonlinear time-history analysis and the target ductility is within an allowable range.

#### 3.2. Ground Motion Data

The main objective of the numerical analysis in this study is the validation of the evaluation procedure presented in the Section 2. As is discussed in the previous study [57], the time-varying function of the momentary energy input of the linear single-degree-of-freedom model is the function of the Fourier amplitude and the phase difference of the ground acceleration. Similarly, for the isotropic linear two-degree-of-freedom model, the time-varying function of the bidirectional momentary energy input is the function of the Fourier amplitude and the phase difference, as is shown in Appendix B. However, it is important to note that the time-history of the acceleration is not uniquely determined from the Fourier amplitude and Fourier phase difference. As stated in [57], a new acceleration can be generated artificially from the original acceleration by shifting the Fourier phase angle: the waveform of the generated acceleration is similar to that of the original acceleration, but its time-history is locally different. In the nonlinear time-history analysis results conducted

by the authors (e.g., [61–63]), groups of artificial ground motions are generated from the same target response spectrum and Fourier phase difference. The only difference in the group of generated ground accelerations was the value of the shifted phase angle. The results show that non-negligible scattering of the peak displacement occurs, even though the response spectrum and Fourier phase difference of the input ground motions are identical. Therefore, the author considers that the local difference of the time-history of the acceleration caused by the shifting phase angle is indispensable to assess the accuracy of the evaluation procedure. Note that the source of uncertainty of the seismic input to a structure is not only the difference of the phase angle of ground motions:, e.g., properties of the fault, direction of rupture, path and site. However, because the main objective of this study is the validation of the evaluation procedure using the time-varying function, only the difference of the phase angle of ground motions is considered in this study: the other source of uncertainty of the seismic input mentioned above is out of the scope of this study.

Based on discussions above, eight ground motion groups are generated from the eight recorded ground motions used in the previous study [58]. The scheme for the generation of artificial ground motions is as follows:

STEP 1: Determine the horizontal major and minor axis according of the horizontal ground motions. Let the $\xi$-axis and $\zeta$-axis be the orthogonal axes in the X–Y plane and consider the following matrix.

$$\mathbf{I} = \begin{bmatrix} I_{A,\xi\xi} & I_{A,\xi\zeta} \\ I_{A,\xi\zeta} & I_{A,\zeta\zeta} \end{bmatrix}, \tag{27}$$

$$\text{where} \begin{cases} I_{A,\xi\xi} = \frac{\pi}{2g} \int\limits_0^{t_d} \{a_{g\xi}(t)\}^2 dt, \ I_{A,\xi\zeta} = \frac{\pi}{2g} \int\limits_0^{t_d} \{a_{g\xi}(t)\}\{a_{g\zeta}(t)\} dt \\ \\ I_{A,\zeta\zeta} = \frac{\pi}{2g} \int\limits_0^{t_d} \{a_{g\zeta}(t)\}^2 dt \end{cases}. \tag{28}$$

In Equation (28), $a_{g\xi}(t)$ and $a_{g\zeta}(t)$ are the $\xi$- and $\zeta$-components of the horizontal ground acceleration, respectively. Following the works done by Arias [64], Penzien and Watabe [65], the horizontal major and minor axes can be obtained as the eigenvectors of the matrix $\mathbf{I}$.

Then, rotate the direction of horizontal excitation with respect to the vertical axis so that the major axis of horizontal excitation coincides with the X-axis.

STEP 2: Calculate the complex Fourier coefficient of the two horizontal components. For the fast Fourier transform (FFT) analysis, zeros were added at the end of each record.

STEP 3: Generate twelve semi-artificial ground motions for each record by shifting the phase angle; the ground acceleration vector $\mathbf{a_g}(t, \Delta\phi_0)$ is expressed as:

$$\mathbf{a_g}(t, \Delta\phi_0) = \sum_{n=-N}^{N} \begin{Bmatrix} c_{X,n} \\ c_{Y,n} \end{Bmatrix} \exp[i\{\omega_n t - \text{sgn}(\omega_n)\Delta\phi_0\}]. \tag{29}$$

where $\Delta\phi_0$ is the constant for shifting the phase angle of all harmonics. In this study, the constant $\Delta\phi_0$ was set from 0 to $11\pi/12$ in intervals of $\pi/12$. Notably, the phase difference of each ground motion component does not change by shifting the phase angle. The generated artificial ground motions are numbered from 00 to 11 depending on $\Delta\phi_0$: the generated ground motions numbered 00 is identical to the original ground motions.

Note that the principal axes of the semi-artificial ground motions remain unchanged from the original records. Detail of the discussions can be found in Appendix C.

Table 1 presents the list of the ground motion groups. Note that some ground motions investigated in the previous study [58] are omitted because the beginning part of such ground motions are missing in the original records due to the late triggering of the old observation system.

**Table 1.** List of ground motion groups investigated in the present study.

| Earthquake of the Original Record | Ground Motion Name of the Original Record | Ground Motion Group ID | Arias Intensity (Original Record) | |
|---|---|---|---|---|
| | | | Major (m/s) | Minor (m/s) |
| a    Hyogo-ken Nanbu, 1995 | JMA Kobe | JKB | 9.640 | 4.201 |
| b    Hokkaido Iburi-Tobu, 2018 | K-Net Mukawa | MKW | 6.797 | 3.984 |
| c    Kumamoto, 2016 (4/16 Earthquake) | KIK-Net Mashiki | MSK | 13.92 | 4.906 |
| d    Northridge, 1994 | Sylmar | SYL | 5.153 | 2.467 |
| e    Chichi, 1999 | TCU075 | TCU | 2.978 | 1.242 |
| f    Kocaeli, 1999 | Yarimka | YPT | 1.432 | 1.221 |
| g    Tokachi-oki, 1968 | Hachinohe [66] | HAC | 1.329 | 1.045 |
| h    Tokachi-Oki, 2003 | K-Net Tomakomai | TOM | 0.517 | 0.449 |

In Table 1, Arias Intensity $I_A$ [64] is calculated from

$$I_{A,X} = \frac{\pi}{2g} \int_0^{t_d} \{a_{gX}(t)\}^2 dt, \, I_{A,Y} = \frac{\pi}{2g} \int_0^{t_d} \{a_{gY}(t)\}^2 dt. \tag{30}$$

In Equation (30), $g$ is the gravity acceleration, $I_{A,X}$ and $I_{A,Y}$ are the Arias Intensity calculated from the as-provided whole record of the major (X) and minor (Y) components, respectively. Ground motion records (a) JMA Kobe (JKB) and (h) Hachinohe (HAC) [66] are chosen because they are widely used for the seismic design of skyscrapers, base-isolated buildings and buildings with various dampers in Japan. Ground Motions (b) to (f) are chosen as the example of so called near-field records. Specifically, ground motion records (e) TCU075 (TCU) and (f) Yarimka (YPT) are chosen as the example of pulse-like ground motions, as investigated by Güneş and Ulucan [67]. Ground Motion record (h) K-Net Tomakomai (TOM) is chosen as the example of long-period long-duration ground motions. The details of the original ground motions can be found in Appendix D. Note that some ground motion records may contain aftershocks (e.g., KIK-Net Mashiki (MSK)) in the records. However, the author thinks this would not affect the main objective of the numerical analysis. Therefore, the as-provided eight ground motions shown in this list are used in this study: no zero is added at the beginning of the original accelerations. The example of the time-history of phase-shifted ground motions are compared in Appendix E.

Figure 6 shows the linear bidirectional pseudo acceleration spectrum with 5 percent of critical damping, calculated from the original and phase-shifted ground motions. The bidirectional pseudo acceleration shown here is calculated from the maximum absolute value of displacement. This figure confirms that the spectrum calculated from the phase-shifted ground acceleration (wave 01 to 11) agrees very well with the spectrum calculated from the original ground motions, as expected.

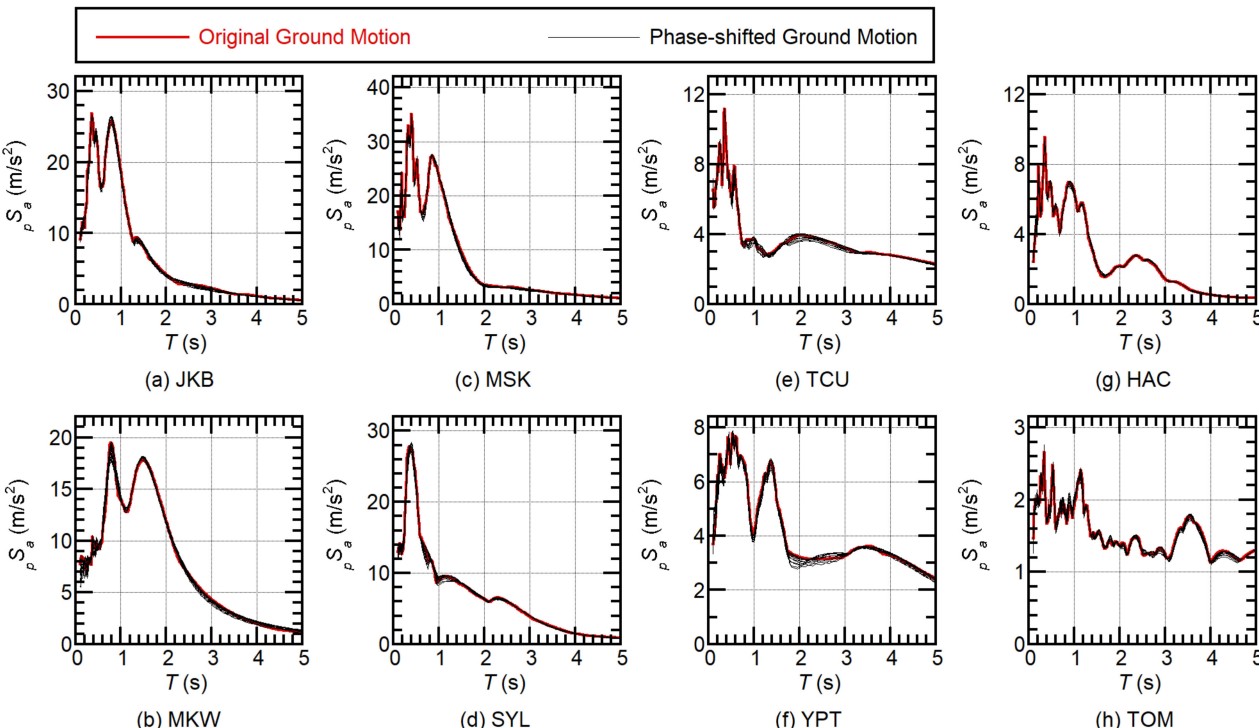

**Figure 6.** Linear bidirectional pseudo-acceleration spectrum calculated from the original and phase-shifted ground motions.

## 4. Analysis Results

First, the evaluated total input energy spectrum ($V_I$ spectrum) and maximum momentary input energy spectrum ($V_{\Delta E}$ spectrum) were compared with the nonlinear time-history analysis results to validate the applicability of the time-varying function. As the accuracy of the time-varying function for a linear system was investigated in a previous study [58], this comparison was conducted for verification in the case of a nonlinear system. Next, the evaluated peak displacement and dimensionless parameter $\gamma$ were compared with the nonlinear time-history analysis results.

### 4.1. Evaluation of Total Input Energy and Maximum Momentary Energy Input

Figure 7 shows the comparison of the $V_I$ spectrum, which was obtained using the time-varying function and nonlinear time-history analysis results. The evaluated $V_I$ spectrum fits well to that obtained from the nonlinear time-history analysis. Additionally, the fluctuation of the nonlinear analysis results is small. Therefore, the $V_I$ spectrum of the nonlinear system can be predicted using the time-varying function, an appropriate effective period $T_{eff}$, and damping $h_{eff} = 0.10$, which is consistent with the results obtained by Akiyama [5].

Figure 8 shows the $V_{\Delta E}$ spectrum comparison. Similarly, the evaluated $V_{\Delta E}$ spectrum is similar to that obtained from the nonlinear time-history analysis results, although there are some fluctuations in the nonlinear analysis results, owing to the local difference in the ground acceleration.

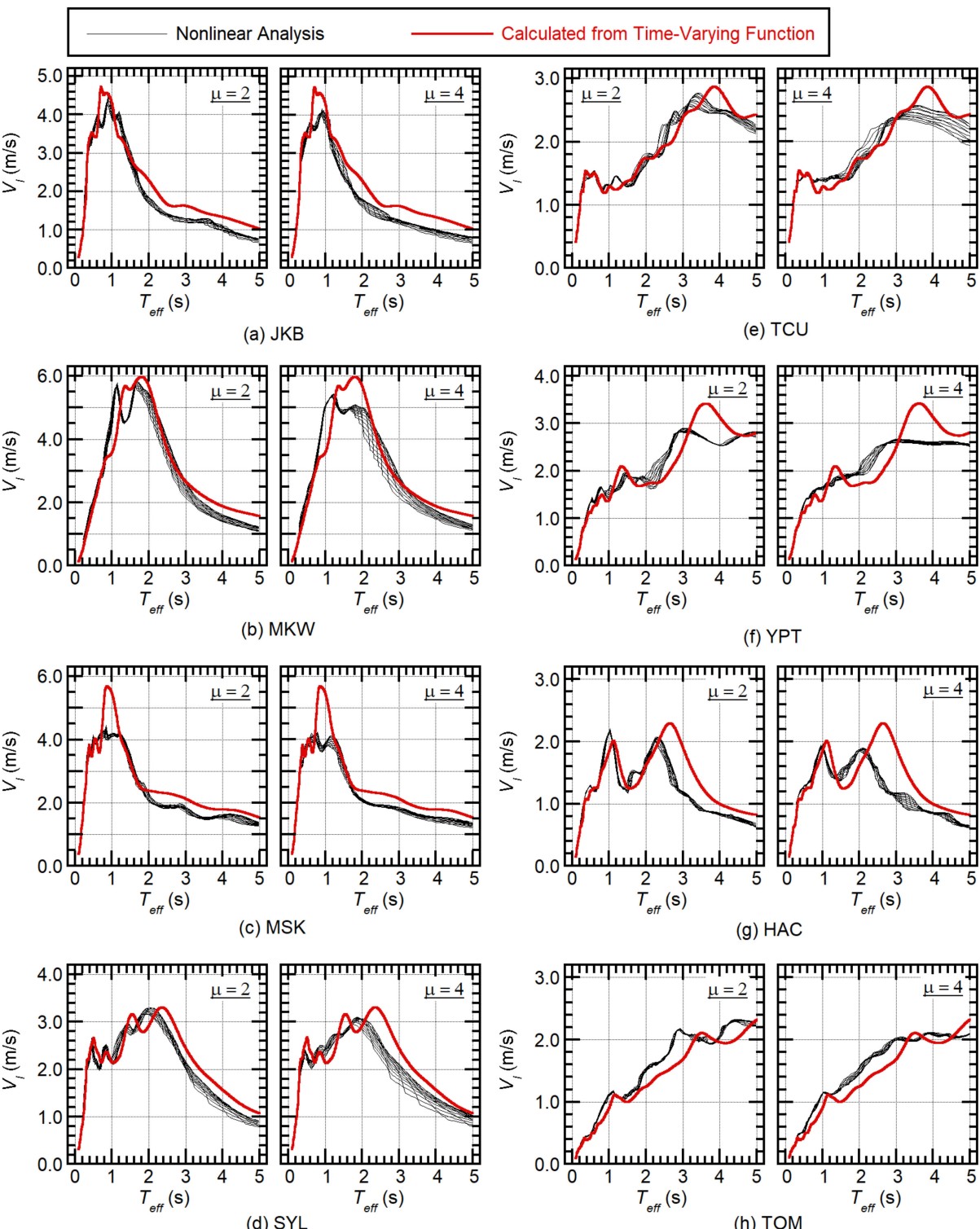

**Figure 7.** Evaluation of total input energy spectrum for each ground acceleration group.

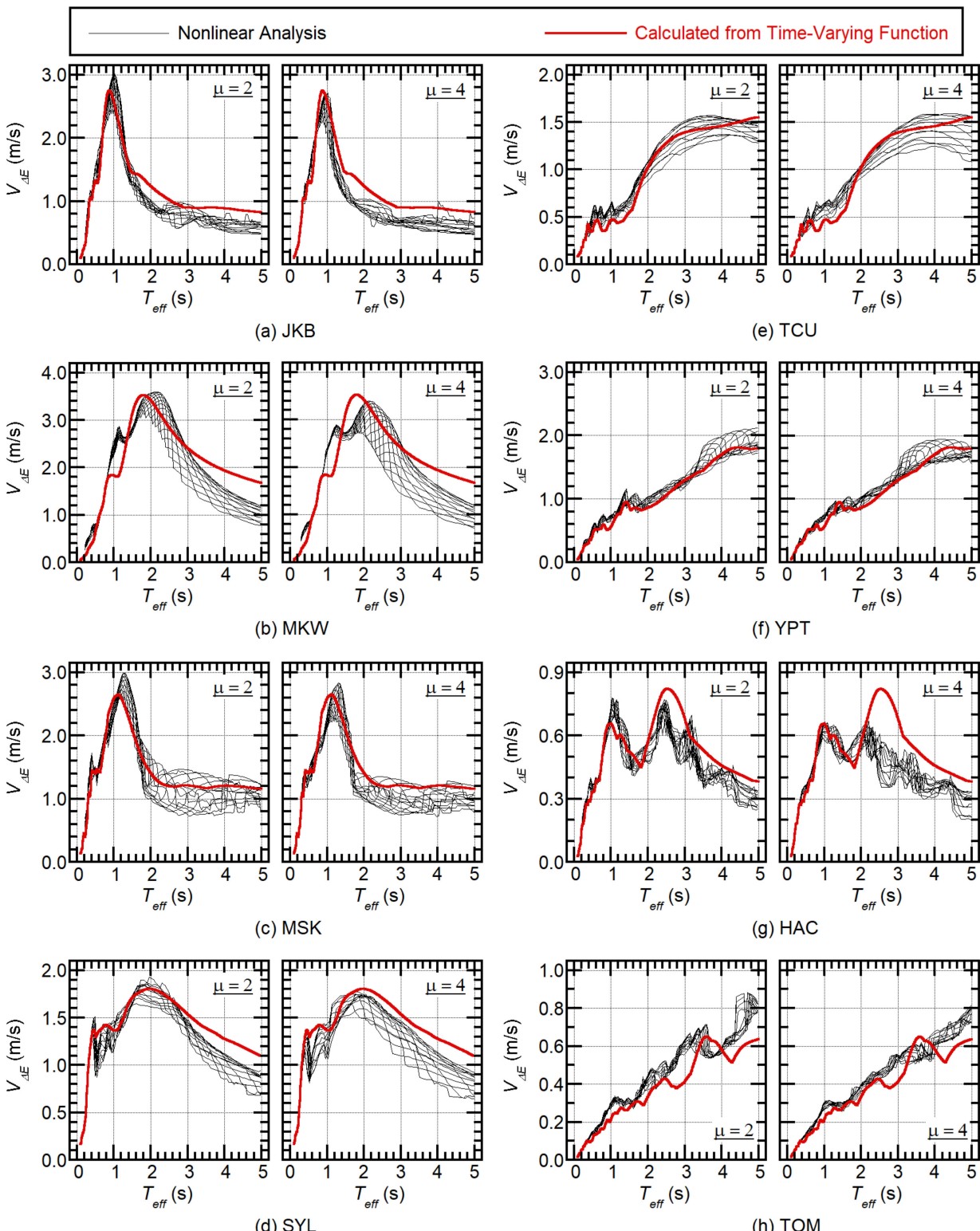

**Figure 8.** Evaluation of maximum momentary input energy spectrum for each ground acceleration group.

*4.2. Evaluation of Nonlinear Peak Displacement and Dimensionless Parameter $\gamma$*

The evaluated nonlinear peak displacement and dimensionless parameter $\gamma$ were compared with the nonlinear analysis results. The evaluated values were calculated using the evaluated $V_I$ and $V_{\Delta E}$ discussed in the previous section.

Figure 9 shows the nonlinear peak displacement evaluated for each ground acceleration group. The figure shows the evaluated results obtained by assuming three hysteresis loop types. The nonlinear analysis results are generally similar to the Type 1 results in the shorter period, and similar to the Type 3 results in the longer period. In most cases, Type 1 provides a conservative estimation. The Type 2 results show the average of the nonlinear results, but the peak displacement in the longer period (for example, (a) JKB) is somewhat overestimated and (h) TOM is underestimated.

Figure 10 shows the evaluated dimensionless parameter $\gamma$ for each ground acceleration group. Most of the nonlinear analysis results are between the Type 1 and the Type 3 results. Type 3 approximates the upper bound of the nonlinear analysis results, while Type 1 gives the lower bound. The Type 2 results are the average of the nonlinear analysis results. Interestingly, in the short period range ($T_{eff} < 1.0$ s), parameter $\gamma$ drastically decreases as $T_{eff}$ increases (for example, (e) TCU), while in the longer period range ($T_{eff} > 1.0$ s), parameter $\gamma$ is approximately constant. Additionally, parameter $\gamma$ tends to increase as the effective duration $t_{D5-95}$ (see Appendix E) becomes longer. In the case of (h) TOM, the lower bound of $\gamma$ in the longer period range is approximately 1.5, while that for the ground motion group shown in (a) to (f) is 0.7.

In conclusion, the peak displacement and $\gamma$ can be evaluated with satisfactory accuracy using the time-varying functions proposed in a previous study [58]. This implies that, for ductile RC structures, the nonlinear peak displacement and cumulative hysteresis energy can be approximately calculated from the properties of the system and the Fourier amplitude and phase difference of the ground acceleration components.

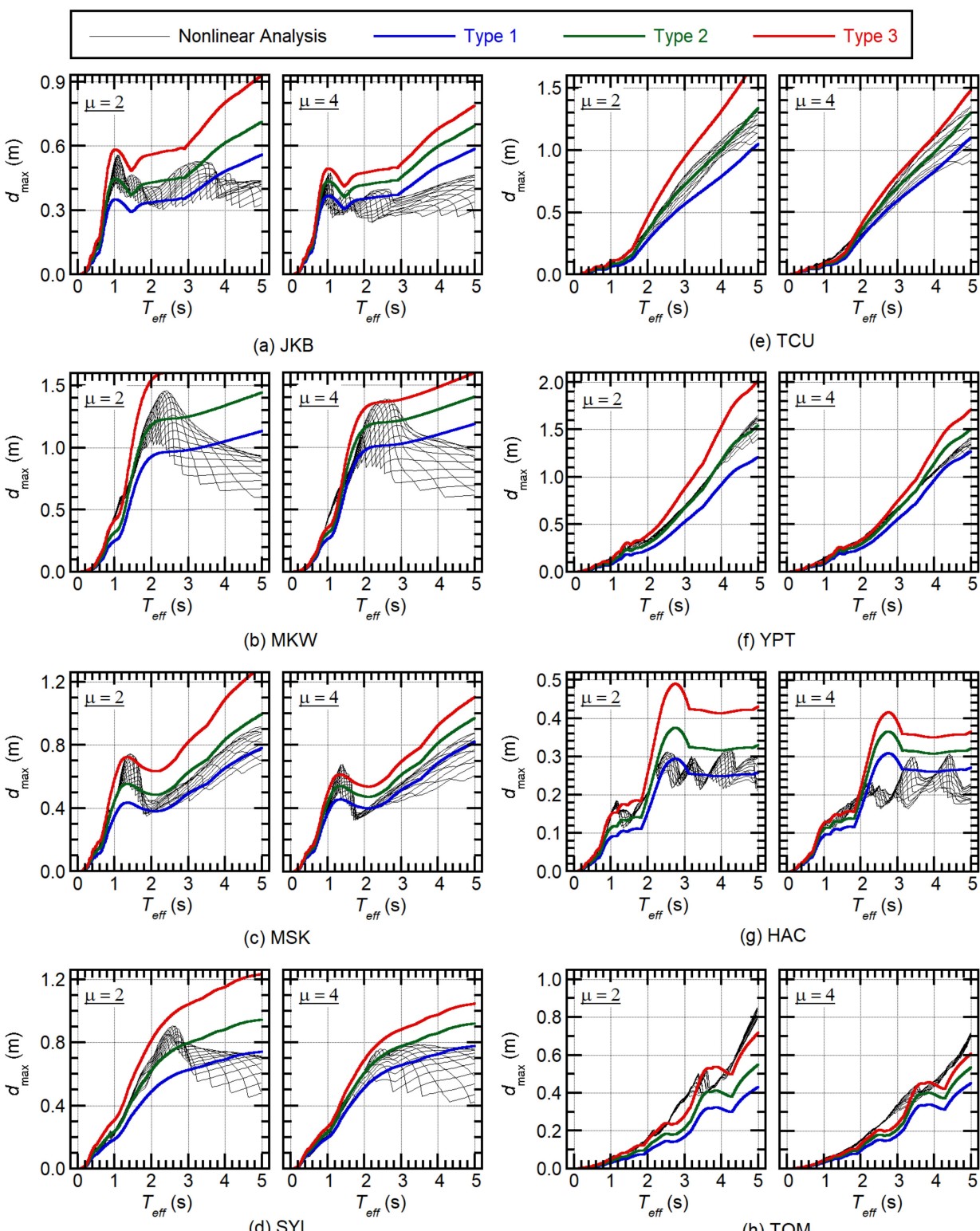

**Figure 9.** Evaluation of nonlinear peak displacement for each ground acceleration group.

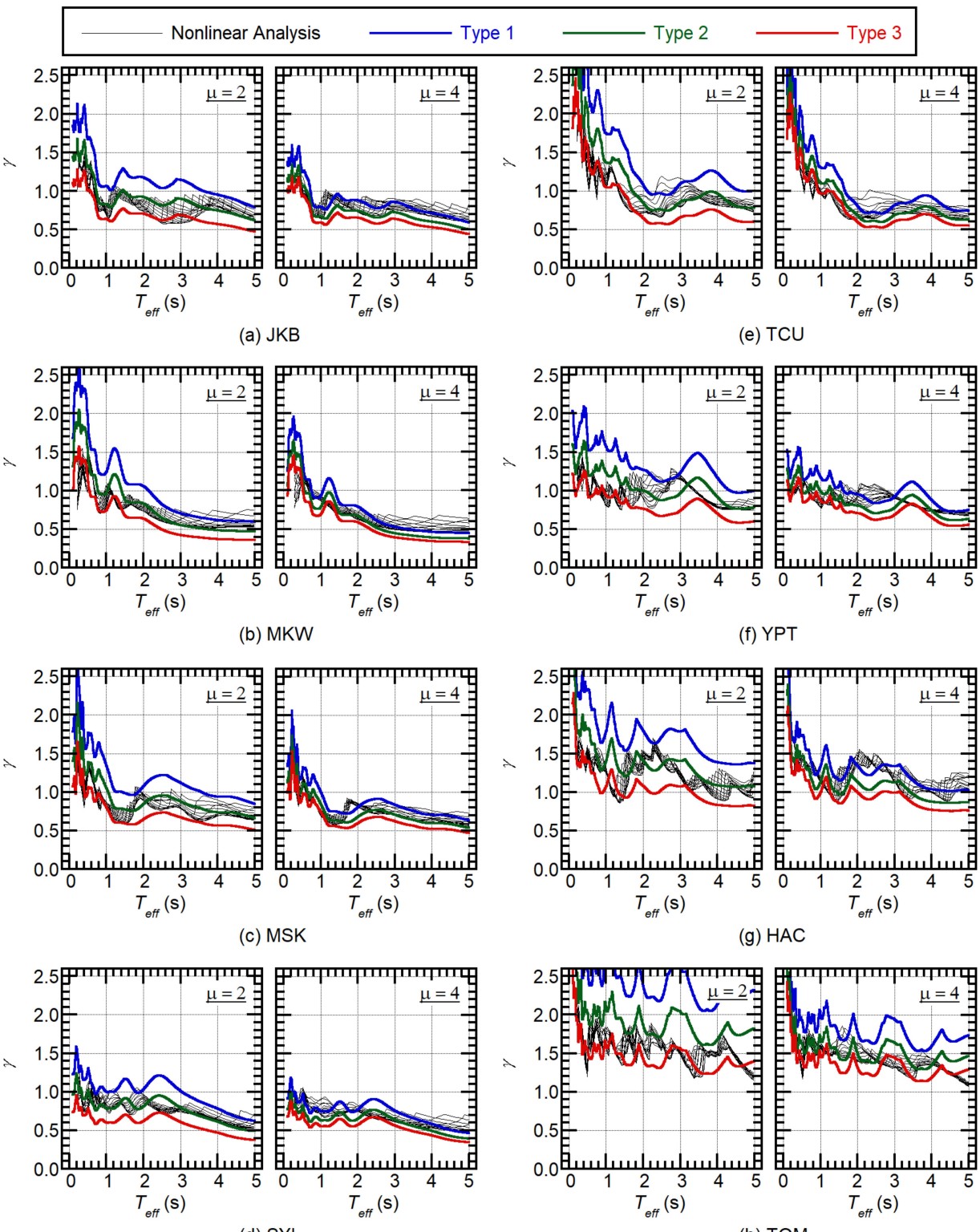

**Figure 10.** Evaluation of dimensionless parameter $\gamma$ for each ground acceleration group.

## 5. Discussion

The following discussion is focused on (i) the validation of the model of the increment in the hysteretic dissipated energy in a half cycle (Types 1 to 3 shown in Figure 4), and (ii) the validation of the formulation of the dimensionless parameter $\gamma$ using the equivalent velocity of the maximum momentary input energy $V_{\Delta E}$ and total input energy $V_I$.

### 5.1. Validation of Model of Increment of Hysteretic Dissipated Energy in Half Cycle

Figure 11 shows the relationship between the $V_I/V_{\Delta E}$ ratio and the $Q_{max}d_{max}/\Delta E_{max}$ ratio. Notably, the tangent stiffness after yielding is negligibly small, and $Q_{max} \approx Q_y$. In this figure, the three horizontal lines representing Types 1 to 3 are shown. For each case, the $Q_{max}d_{max}/\Delta E_{max}$ ratio is calculated as follows:

$$\frac{Q_{max}d_{max}}{\Delta E_{max}} \approx (0.85)^2 \frac{\mu}{f(\mu)}. \tag{31}$$

The deviation of Equation (31) can be found in Appendix F. Figure 11a shows that, as expected, most plots of the nonlinear analysis results are between Type 1 and 3 for $\mu = 2$. Additionally, the $Q_{max}d_{max}/\Delta E_{max}$ ratio tends to increase with $V_I/V_{\Delta E}$. A similar observation was made for $\mu = 4$, as shown in Figure 11b, but more scatters were observed compared with Figure 11a.

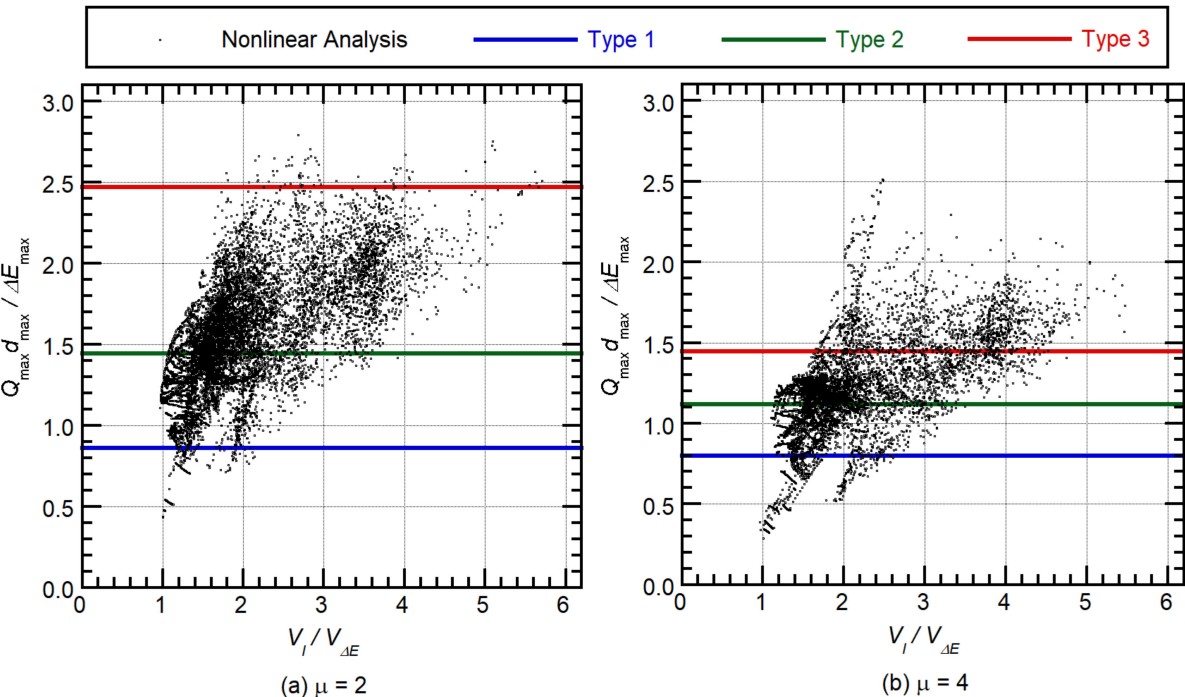

**Figure 11.** Relationship between $V_I/V_{\Delta E}$ and $Q_{max}d_{max}/\Delta E_{max}$ obtained from nonlinear time-history analysis.

Figure 12 shows the relationship between the effective period $T_{eff}$ and the $Q_{max}d_{max}/\Delta E_{max}$ ratio for both $\mu = 2$ and $\mu = 4$. Although large scatter can be observed in the nonlinear analysis plots, $Q_{max}d_{max}/\Delta E_{max}$ slightly decreases as $T_{eff}$ increases. This trend is consistent with the results shown in Figure 9.

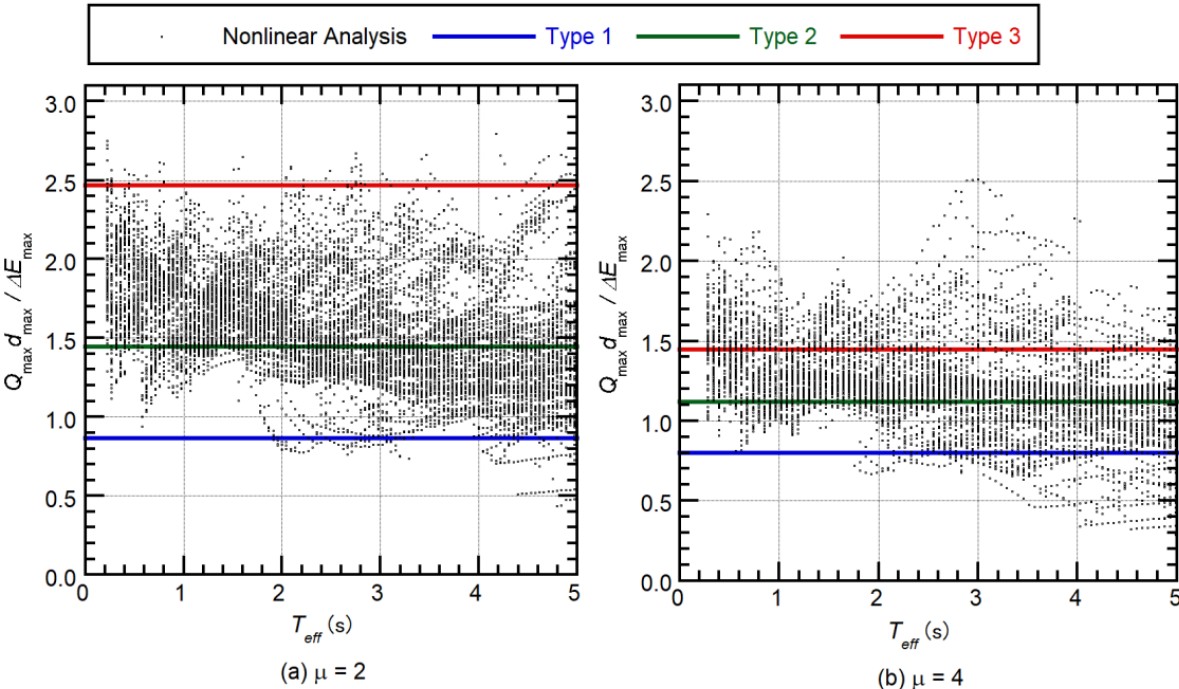

**Figure 12.** Relationship between effective period $T_{eff}$ and $Q_y d_{max}/\Delta E_{max}$ obtained from nonlinear analysis results.

In conclusion, to evaluate the nonlinear peak displacement, the model of the increment in the hysteresis dissipated energy in a half cycle, which was proposed by Nakamura et al. [53] for unidirectional excitation, may be used in the case of $2 \leq \mu \leq 4$. Notably, the large fluctuation of $Q_{max}d_{max}/\Delta E_{max}$, which is caused by the local difference in the ground acceleration, is unavoidable.

*5.2. Validation of Formulation of Fajfar's Parameter $\gamma$ Using Maximum Momentary Input Energy and Total Input Energy*

Figure 13 shows the comparison of the relationship between $V_I/V_{\Delta E}$ and the dimensionless parameter $\gamma$, which was obtained from the nonlinear analysis results and Equation (18) for three cases (Types 1 to 3). Most plots of the nonlinear analysis results are between Type 1 and 3. Therefore, the $\gamma$ evaluated using Equation (18) is satisfactorily accurate, provided that the $V_I/V_{\Delta E}$ ratio is accurately evaluated and the hysteretic dissipated energy in a half cycle is appropriately assumed.

Figure 14 shows the evaluation of $V_I/V_{\Delta E}$ for each ground acceleration group. As shown in this figure, the evaluated $V_I/V_{\Delta E}$ is in good agreement with the nonlinear analysis results, and the fluctuation in the nonlinear analysis results is relatively small. Therefore, the evaluation of the $V_I/V_{\Delta E}$ ratio using a time-varying function (Equations (23)–(25)) has satisfactory accuracy.

Figure 14 also shows that $V_I/V_{\Delta E}$ decreases as the effective period $T_{eff}$ increases, which implies that cyclic loading exerts significant influence on the structures in the shorter period range, because the $V_I/V_{\Delta E}$ ratio indicates the number of cyclic loads.

In conclusion, to evaluate the dimensionless parameter $\gamma$, Equation (18) can be used in the case of $2 \leq \mu \leq 4$. The evaluated ratio $V_I/V_{\Delta E}$ for a given effective period $T_{eff}$ using a time-varying function is sufficiently accurate for evaluating $\gamma$. The main reason for the scatter in the evaluation of $\gamma$ seems to be the large fluctuation of $Q_{max}d_{max}/\Delta E_{max}$.

*5.3. Comparisons with the Fajfar's Formulation of Parameter $\gamma$*

In this subsection, an empirical formula proposed by Fajfar and Vidic [13] is discussed in based on the results above. Since the empirical formula in reference [13] is not for the bidirectional excitation, direct comparisons of the numerical aspect are difficult. Therefore,

the following discussion is made focusing on what parameters affect the parameter $\gamma$, from the comparison of equations.

The empirical formula proposed by Fajfar and Vidic [13] is

$$\gamma = z_T z_\mu z_g. \tag{32}$$

In Equation (32), $z_T$, $z_\mu$, and $z_g$ are functions of the natural period $T$, ductility and ground motion. Fajfar and Vidic [13] have formulated function $z_T$ for the elastic spectrum of the Newmark–Hall type which depends on the hysteresis model: according to Fajfar and Vidic [13], $z_T$ for the ductile reinforced concrete structures ("Q-model" in Fajfar and Vidic) is

$$z_T = \begin{cases} 1.05 - 0.3\frac{T}{T_1} & : T \le T_1 \\ 0.75 - 0.25\frac{T-T_1}{T_2-T_1} & : T_1 \le T \le T_2 \\ 0.50 & : T_2 \le T \end{cases} . \tag{33}$$

where $T_1$ and $T_2$ are the transition function periods which represent the limit between the short-, medium- and long-period region of the Newmark–Hall elastic spectrum. Similarly, functions $z_\mu$ and $z_g$ proposed in Fajfar and Vidic [13] for the ductile reinforced concrete structures are

$$z_\mu = \frac{(\mu - 1)^{0.58}}{\mu}, \tag{34}$$

$$z_g = \left[ \frac{1}{a_{g1max} v_{g1max}} \int_0^{t_d} \{a_{g1}(t)\}^2 dt \right]^{0.4}. \tag{35}$$

where $a_{g1max}$ and $v_{g1max}$ are the peak ground acceleration and peak velocity of the ground acceleration, respectively, and $a_{g1}(t)$ is the time-history of ground acceleration.

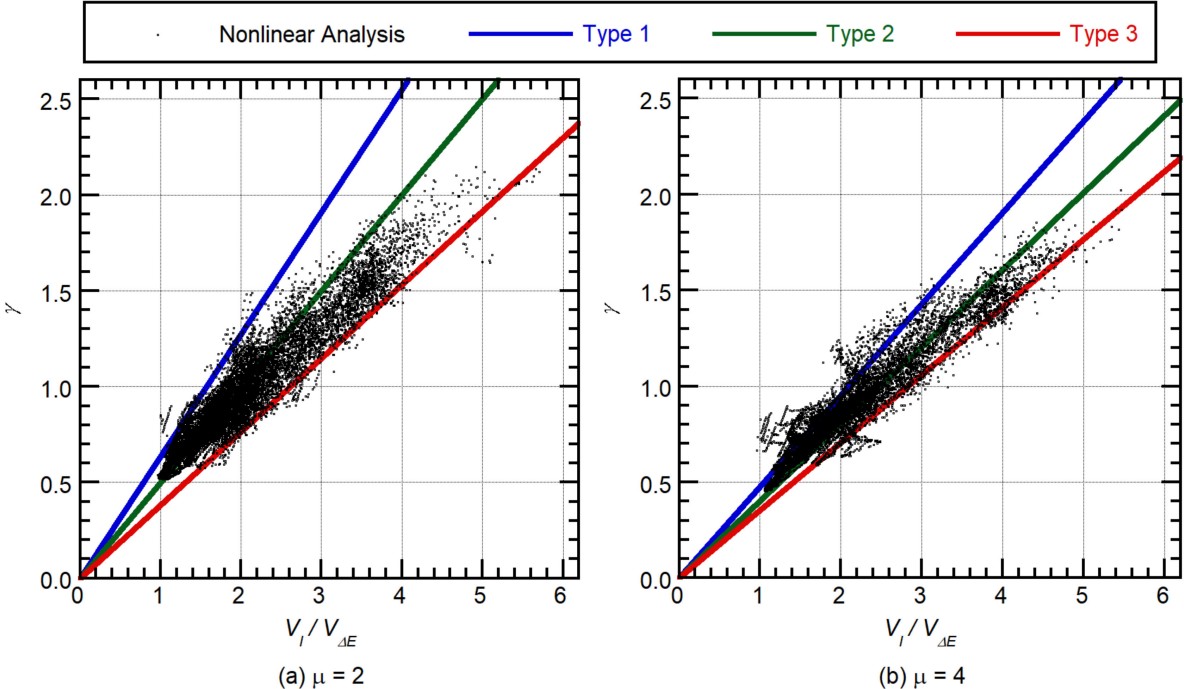

**Figure 13.** Comparison of relationship between $V_I/V_{\Delta E}$ and $\gamma$ obtained from nonlinear analysis results and equations considering three types of modelled hysteretic dissipated energy in a half cycle.

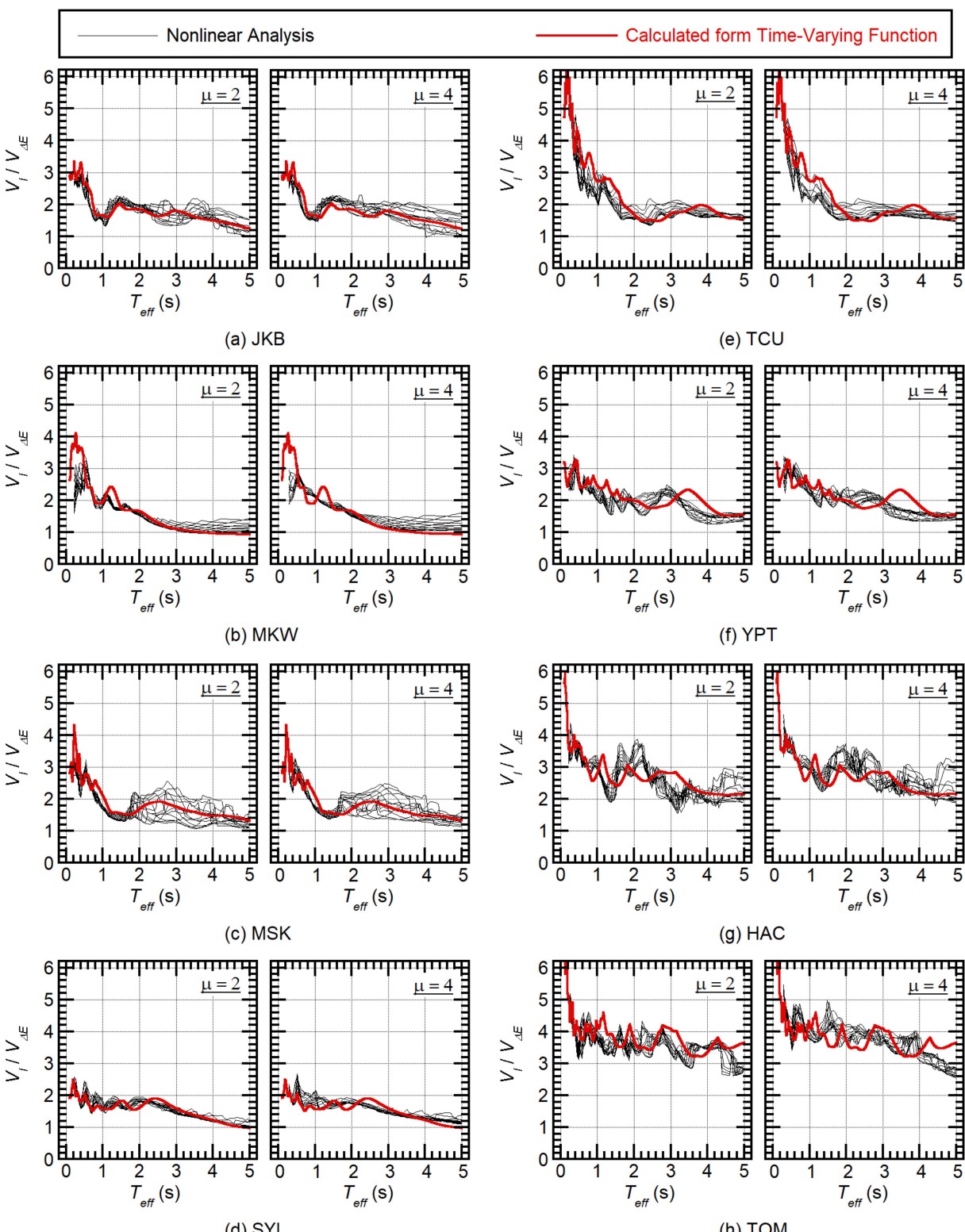

**Figure 14.** Evaluation of the $V_I / V_{\Delta E}$ ratio for each ground acceleration group.

From the comparisons between Equation (18) and Equations (32)–(35), the following observations can be made:

- The ratio $V_I/V_{\Delta E}$ in Equation (18) corresponds to the product $z_T z_g$ in Equations (33) and (35). As is observed in Figure 14, the ratio $V_I/V_{\Delta E}$ decreases as $T$ increases, while $V_I/V_{\Delta E}$ is larger for the long-duration ground motions (e.g., (g) HAC and (h) TOM in Figure 14).
- The ratio $\sqrt{f(\mu)}/\mu$ corresponds to $z_\mu$ formulated in Equation (34).

Based on the observations, the proposed equation in this study (Equation (18)) agrees with the empirical equation proposed by Fajfar and Vidic [13] qualitatively. Note that Equation (18) has a theoretical background (energy balance during a half cycle of structural response) and is applicable to any hysteresis model, provided the function $f(\mu)$ is properly formulated from the hysteresis loop. In addition, the ratio $V_I/V_{\Delta E}$ can be calculated from the time-varying function of the momentary energy input discussed in this study. Therefore, the proposed equation in this study (Equation (18)) is the generalized version of Equations (32)–(35).

## 6. Conclusions

This study evaluated the bidirectional seismic input to an isotropic nonlinear one-mass two-degree-of-freedom system. The main conclusions and results are as follows:

- The maximum momentary input spectrum and total input energy spectrum of a nonlinear system can be evaluated using a time-varying function, an appropriate effective period, and the damping ratio. This is consistent with unidirectional excitation.
- The nonlinear peak displacement of an isotropic nonlinear system can be evaluated with satisfactory accuracy by using the maximum momentary input energy evaluated by the time-varying function. For unidirectional excitation, the model of the increment in the hysteresis-dissipated energy in a half cycle, which was proposed by Nakamura et al. [53], may be used to evaluate the nonlinear peak displacement.
- A simple equation is proposed to calculate parameter $\gamma$ from the maximum momentary input energy and total input energy. The numerical analysis results reveal that the $\gamma$ evaluated using the proposed equation is in good agreement with the nonlinear time-history analysis results.

The bidirectional momentary input energy investigated herein has the following advantages: (i) it can be directly calculated from the Fourier amplitude and phase angle of the ground motion components using the time-varying function of the momentary energy input, without knowing the time-history of the ground motion; (ii) it is easy to apply to nonlinear systems for the evaluation of the peak displacement and cumulative hysteresis energy. Advantage (i) is most important because it means that researchers can eliminate otherwise unavoidable fluctuations from the nonlinear time-history analysis results.

Notably, there are unavoidable fluctuations in the evaluation of the peak displacement and parameter $\gamma$. To conservatively evaluate the peak displacement, a Type 3 model (smallest normalized cumulative hysteresis energy increment during a half cycle) may be used. However, in this case, the evaluation of parameter $\gamma$ will not be conservative. Therefore, the author thinks that the appropriate selection of the assumed model of the hysteresis loop during a half cycle depends on the properties of the structure and the damage model applied to the structure. Let us consider the case wherein the Park–Ang model (Equation (11)) is used for damage assessment. If the parameter $\beta$ is large (the damage caused by cumulative energy is predominant), the assumed model should be selected such that parameter $\gamma$ is conservatively evaluated. However, if $\beta$ is small (the damage caused by the peak deformation is predominant), the assumed model should be selected such that the peak displacement is conservatively evaluated.

Future work should consider the application of the bidirectional momentary energy input spectrum to (a) the prediction of the largest peak displacement of multi-story irregular buildings subjected to horizontal bidirectional seismic input, and (b) base-isolated structures subjected to horizontal bidirectional seismic input. According to the code-specific ground motions (mostly defined by the acceleration spectrum), the relationship between

the equivalent velocity of the maximum momentary input energy and the spectral velocity should also be investigated.

**Author Contributions:** All contributions related to this article were made by the first (single) author, except for the English language editing, as mentioned in the acknowledgements. All authors have read and agreed to the published version of the manuscript.

**Funding:** This study did not receive external funding.

**Institutional Review Board Statement:** Not applicable.

**Informed Consent Statement:** Not applicable.

**Data Availability Statement:** The data presented in this study are available on request from the corresponding author. The data are not publicly available because they are not part of ongoing research.

**Acknowledgments:** The ground motions used in this study were obtained from the website of the Japan Meteorological Agency (https://www.data.jma.go.jp/svd/eqev/data/kyoshin/index.htm, accessed on 14 December 2019), National Research Institute for Earth Science and Disaster Resilience (NIED) (http://www.kyoshin.bosai.go.jp/kyoshin/, accessed on 14 December 2019), and Pacific Earthquake Engineering Research Center (PEER) (https://ngawest2.berkeley.edu/, accessed on 14 December 2019).

**Conflicts of Interest:** The author declares that there is no conflict of interest.

## Appendix A. Time-Varying Function of Bidirectional Momentary Energy Input to an Isotropic Linear Two-Degree-Of-Freedom System

The time-varying function of the ground acceleration and energy input [57,58] is summarized below. The discrete time-history of the ground acceleration vector $\mathbf{a_g}(t)$, which is defined within the range $[0, t_d]$, can be expressed as a Fourier series, as follows:

$$\mathbf{a_g}(t) = \left\{ \begin{array}{c} a_{gX}(t) \\ a_{gY}(t) \end{array} \right\} = \sum_{n=-N}^{N} \left\{ \begin{array}{c} c_{X,n} \\ c_{Y,n} \end{array} \right\} \exp(i\omega_n t), \tag{A1}$$

$$\text{where } \omega_n = n\Delta\omega = n(2\pi/t_d). \tag{A2}$$

In Equation (A1), the coefficients $c_{X,n}$ and $c_{Y,n}$ are the $n$th complex Fourier coefficient of the X- and Y-components of the ground acceleration, respectively; $\omega_n$ is the circular frequency of the $n$th harmonic. It is assumed that both $c_{X,0}$ and $c_{Y,0}$ are zero. Similar to $\mathbf{a_g}(t)$, another ground motion $\mathbf{a_g}^*(t)$ is defined as:

$$\mathbf{a_g}^*(t) = \left\{ \begin{array}{c} a_{gX}^*(t) \\ a_{gY}^*(t) \end{array} \right\} = -i \sum_{n=-N}^{N} \left\{ \begin{array}{c} c_{X,n} \\ c_{Y,n} \end{array} \right\} \mathrm{sgn}(\omega_n) \exp(i\omega_n t), \tag{A3}$$

$$\text{where } \mathrm{sgn}(\omega_n) = \begin{cases} 1 & : \omega_n > 0 \\ -1 & : \omega_n < 0 \end{cases}. \tag{A4}$$

According to a previous study [57], the envelope function of the horizontal ground motion is defined as:

$$\{\alpha(t)\}^2 = \{\alpha_X(t)\}^2 + \{\alpha_Y(t)\}^2, \tag{A5}$$

$$\text{where } \begin{cases} \alpha_X(t) = \sqrt{\{a_{gX}(t)\}^2 + \{a_{gX}^*(t)\}^2} \\ \alpha_Y(t) = \sqrt{\{a_{gY}(t)\}^2 + \{a_{gY}^*(t)\}^2} \end{cases}. \tag{A6}$$

As shown in [57], the envelope function of the X- and Y-components, $\alpha_X(t)$ and $\alpha_Y(t)$, can be, respectively, expressed in form of a Fourier series, as follows:

$$\begin{cases} \{\alpha_X(t)\}^2 = 2 \sum\limits_{n=-N+1}^{N-1} {}_2A_{GX,n}{}^* \exp(i\omega_n t) \\ \{\alpha_Y(t)\}^2 = 2 \sum\limits_{n=-N+1}^{N-1} {}_2A_{GY,n}{}^* \exp(i\omega_n t) \end{cases}, \tag{A7}$$

$$\text{where} \begin{cases} {}_2A_{GX,n}{}^* = \begin{cases} 2 \sum\limits_{n_1=n+1}^{N} c_{X,n_1} c_{X,-(n_1-n)} & : n \geq 0 \\ \overline{{}_2A_{GX,-n}{}^*} & : n < 0 \end{cases} \\ {}_2A_{GY,n}{}^* = \begin{cases} 2 \sum\limits_{n_1=n+1}^{N} c_{Y,n_1} c_{Y,-(n_1-n)} & : n \geq 0 \\ \overline{{}_2A_{GY,-n}{}^*} & : n < 0 \end{cases} \end{cases}. \tag{A8}$$

Notably, the bar over the symbol indicates a complex conjugate. By substituting Equation (A7) into Equation (A5), the bidirectional envelope function of the horizontal ground motion can be expressed as a Fourier series, as follows:

$$\{\alpha(t)\}^2 = 2 \sum_{n=-N+1}^{N-1} {}_2A_{G,n}{}^* \exp(i\omega_n t). \tag{A9}$$

The complex Fourier coefficient of Equation (A9) can be calculated as follows:

$$_2A_{G,n}{}^* = \begin{cases} 2 \sum\limits_{n_1=n+1}^{N} \left\{ c_{X,n_1} c_{X,-(n_1-n)} + c_{Y,n_1} c_{Y,-(n_1-n)} \right\} & : n \geq 0 \\ \overline{{}_2A_{G,-n}{}^*} & : n < 0 \end{cases}. \tag{A10}$$

Let us consider the response of an isotropic linear two-degree-of-freedom system (natural circular frequency, $\omega_0$, damping ratio $h$). According to a previous study [58], the time-varying function of the input energy ratio per unit mass to an isotropic linear two-degree-of-freedom system, $\widehat{e_{I,BI}}$, is defined as:

$$\widehat{e_{I,BI}} = -\frac{1}{2} \left\{ \dot{\mathbf{d}}(t)^{\mathbf{T}} \mathbf{a_g}(t) + \dot{\mathbf{d}}^*(t)^{\mathbf{T}} \mathbf{a_g}^*(t) \right\}, \tag{A11}$$

$$\begin{cases} \dot{\mathbf{d}}(t) = -\sum\limits_{n=-N}^{N} \begin{Bmatrix} c_{X,n} \\ c_{Y,n} \end{Bmatrix} H_V(i\omega_n) \exp(i\omega_n t) \\ \dot{\mathbf{d}}^*(t) = i \sum\limits_{n=-N}^{N} \begin{Bmatrix} c_{X,n} \\ c_{Y,n} \end{Bmatrix} H_V(i\omega_n) \text{sgn}(\omega_n) \exp(i\omega_n t) \end{cases}, \tag{A12}$$

$$H_V(i\omega_n) = i\omega_n H_D(i\omega_n), H_D(i\omega_n) = \frac{1}{\omega_0^2 - \omega_n^2 + 2h\omega_n\omega_0 i}. \tag{A13}$$

By substituting Equations (A1), (A3), and (A12) into Equation (A11), the following relationship can be obtained:

$$\hat{e_{I,BI}} = \sum_{n=-N+1}^{N-1} E_{BI,n}{}^* \exp(i\omega_n t), \tag{A14}$$

$$\text{where } E_{BI,n}{}^* = \begin{cases} \sum\limits_{n_1=n+1}^{N} \{H_V(i\omega_{n_1}) + H_V(-i\omega_{n_1-n})\} \left\{ c_{X,n_1} c_{X,-(n_1-n)} + c_{Y,n_1} c_{Y,-(n_1-n)} \right\} \\ : n \geq 0 \\ \overline{E_{BI,n}{}^*} \\ : n < 0 \end{cases}. \tag{A15}$$

The average of the momentary input energy ratio during time $\Delta t$ per unit mass is approximated as:

$$\frac{1}{\Delta t}\frac{\Delta E_{BI}(t)}{m} \approx \frac{1}{\Delta t}\frac{\Delta \hat{E}_{BI}(t)}{m} = \int_{t-\Delta t/2}^{t+\Delta t/2} e_{I,BI}\hat{}dt = \sum_{n=-N+1}^{N-1} E_{\Delta BI,n}{}^* \exp(i\omega_n t), \quad \text{(A16)}$$

$$\text{where } E_{\Delta BI,n}{}^* = \begin{cases} \frac{\sin(\omega_n \Delta t/2)}{\omega_n \Delta t/2} E_{BI,n}{}^* & : n > 0 \\ E_{BI,0}{}^* = 2\sum_{n_1=1}^{N} \text{Re}\{H_V(i\omega_{n_1})\}\{|c_{X,n_1}|^2 + |c_{Y,n_1}|^2\} & : n = 0 \\ \overline{E_{\Delta BI,n}{}^*} & : n < 0 \end{cases} . \quad \text{(A17)}$$

The calculation of Equations (A16) and (A17) assumes that the duration of a half cycle of response $\Delta t$ can be approximated as half of the response period $T'$, as follows:

$$\Delta t \approx \frac{T'}{2} = \pi \sqrt{\sum_{n=1}^{N} |H_D(i\omega_n)|^2 \{|c_{X,n}|^2 + |c_{Y,n}|^2\} / \sum_{n=1}^{N} |H_V(i\omega_n)|^2 \{|c_{X,n}|^2 + |c_{Y,n}|^2\}}. \quad \text{(A18)}$$

Equation (A16) is the time-varying function of the momentary input energy. The momentary input energy per unit mass at time $t$ can be calculated as follows:

$$\frac{\Delta E_{BI}(t)}{m} \approx \int_{t-\Delta t/2}^{t+\Delta t/2} \frac{1}{\Delta t}\frac{\Delta \hat{E}_{BI}(t)}{m}dt = \int_{t-\Delta t/2}^{t+\Delta t/2} \sum_{n=-N+1}^{N-1} E_{\Delta BI,n}{}^* \exp(i\omega_n t)dt. \quad \text{(A19)}$$

The total input energy per unit mass can be calculated as follows:

$$\frac{E_I}{m} = \int_{0}^{t_d} \sum_{n=-N+1}^{N-1} E_{\Delta BI,n}{}^* \exp(i\omega_n t)dt = t_d E_{BI,0}{}^*. \quad \text{(A20)}$$

Notably, the total input energy per unit mass calculated from (A20) is the *exact* value, and the result is identical to the formulation of Ordaz et al. [68]. Conversely, the momentary input energy per unit mass at time $t$ calculated from Equation (A19) is the *approximate* value.

### Appendix B. Influence of the Phase Angle on the Time-Varying Function of Bidirectional Momentary Energy Input

In Appendix B, the influence of the phase angle on the time-varying function of bidirectional momentary energy input is discussed as in the previous study [57]. Let $A_{X,n}$, $A_{Y,n}$ and $\phi_{X,n}$, $\phi_{Y,n}$ be the $n$th Fourier amplitude and phase angle of the X- and Y-components of the ground acceleration, respectively. The relation between $A_{X,n}$, $A_{Y,n}$, $\phi_{X,n}$, $\phi_{Y,n}$ and $c_{X,n}$, $c_{Y,n}$ is

$$\begin{cases} |c_{X,n}| = |c_{X,-n}| = \frac{A_{X,n}}{2}, c_{X,n} = \frac{A_{X,n}}{2}\exp(-i\phi_{X,n}), c_{X,-n} = \overline{c_{X,n}} = \frac{A_{X,n}}{2}\exp(i\phi_{X,n}) \\ |c_{Y,n}| = |c_{Y,-n}| = \frac{A_{Y,n}}{2}, c_{Y,n} = \frac{A_{Y,n}}{2}\exp(-i\phi_{Y,n}), c_{Y,-n} = \overline{c_{Y,n}} = \frac{A_{Y,n}}{2}\exp(i\phi_{Y,n}) \end{cases} . \quad \text{(A21)}$$

Then, the coefficient $E_{BI,n}{}^*$ shown in Equation (A15) is rewritten using $A_{X,n}$, $A_{Y,n}$, $\phi_{X,n}$, and $\phi_{Y,n}$ as follows. In case of $1 \leq n \leq N-1$, $E_{BI,n}{}^*$ is rewritten as

$$\begin{aligned} E_{BI,n}{}^* = \ &\frac{1}{4}\sum_{n_1=n+1}^{N} \{H_V(i\omega_{n_1}) + H_V(-i\omega_{n_1-n})\} A_{X,n_1} A_{X,n_1-n} \exp\{-i(\phi_{X,n_1} - \phi_{X,n_1-n})\} \\ &+\frac{1}{4}\sum_{n_1=n+1}^{N} \{H_V(i\omega_{n_1}) + H_V(-i\omega_{n_1-n})\} A_{Y,n_1} A_{Y,n_1-n} \exp\{-i(\phi_{Y,n_1} - \phi_{Y,n_1-n})\} \end{aligned} \quad \text{(A22)}$$

The $n$th phase difference of X- and Y-components is defined as

$$\Delta\phi_{X,n} = \phi_{X,n+1} - \phi_{X,n}, \Delta\phi_{Y,n} = \phi_{Y,n+1} - \phi_{Y,n}. \quad \text{(A23)}$$

The difference between the phase angles in Equation (A22) can be rewritten as

$$
\begin{cases}
\phi_{X,n_1} - \phi_{X,n_1-n} = \sum_{n_2=0}^{n-1} \left( \phi_{X,n_1-n_2} - \phi_{X,n_1-n_2-1} \right) = \sum_{n_2=0}^{n-1} \Delta\phi_{X,n_1-n_2-1} \\
\phi_{Y,n_1} - \phi_{Y,n_1-n} = \sum_{n_2=0}^{n-1} \left( \phi_{Y,n_1-n_2} - \phi_{Y,n_1-n_2-1} \right) = \sum_{n_2=0}^{n-1} \Delta\phi_{Y,n_1-n_2-1}
\end{cases}
\quad \text{(A24)}
$$

By substituting Equation (A24) into Equation (A22), the coefficient $E_{BI,n}{}^*$ can be rewritten as

$$
\begin{aligned}
E_{BI,n}{}^* &= \frac{1}{4} \sum_{n_1=n+1}^{N} \left\{ H_V(i\omega_{n_1}) + H_V(-i\omega_{n_1-n}) \right\} A_{X,n_1} A_{X,n_1-n} \exp\left\{ -i\left( \sum_{n_2=0}^{n-1} \Delta\phi_{X,n_1-n_2-1} \right) \right\} \\
&+ \frac{1}{4} \sum_{n_1=n+1}^{N} \left\{ H_V(i\omega_{n_1}) + H_V(-i\omega_{n_1-n}) \right\} A_{Y,n_1} A_{Y,n_1-n} \exp\left\{ -i\left( \sum_{n_2=0}^{n-1} \Delta\phi_{Y,n_1-n_2-1} \right) \right\}
\end{aligned}
\quad \text{(A25)}
$$

Equation (A25) indicates that the coefficient $E_{BI,n}{}^*$ is a function of the *k*th Fourier amplitude $A_{X,k}$, $A_{Y,k}$ and the *k*th phase difference $\Delta\phi_{X,k}$, $\Delta\phi_{Y,k}$. Thus, the time-varying function of bidirectional momentary energy input remains unchanged if the phase angle in the harmonic ground motion is shifted by a constant.

## Appendix C. The Horizontal Principal Axis of the Phase-Shifted Ground Acceleration Components

In Appendix C, the horizontal principal axis of the phase-shifted ground acceleration components is discussed. Consider the matrix calculating the horizontal ground motion components

$$
\mathbf{I} = \frac{\pi}{2g}
\begin{bmatrix}
\int_0^{t_d} \{a_{gX}(t)\}^2 dt & \int_0^{t_d} \{a_{gX}(t)\}\{a_{gY}(t)\} dt \\
\int_0^{t_d} \{a_{gX}(t)\}\{a_{gY}(t)\} dt & \int_0^{t_d} \{a_{gY}(t)\}^2 dt
\end{bmatrix}.
\quad \text{(A26)}
$$

From the definition of the principal axis of the horizontal ground acceleration by Arias [64], Penzien and Watabe [65], the non-diagonal terms in (A26) are zero when the X-axis coincides with the horizontal major axis, i.e.,

$$
\int_0^{t_d} \{a_{gX}(t)\}\{a_{gY}(t)\} dt = 0.
\quad \text{(A27)}
$$

Equation (A27) can be rewritten as

$$
\int_0^{t_d} \{a_{gX}(t)\}\{a_{gY}(t)\} dt = t_d \sum_{n=-N+1}^{N-1} c_{X,n} c_{Y,-n}.
\quad \text{(A28)}
$$

Therefore, the following relation is obtained between the complex Fourier coefficients of the two components as

$$
\sum_{n=-N+1}^{N-1} c_{X,n} c_{Y,-n} = 0.
\quad \text{(A29)}
$$

Next, the principal axis of the phase-shifted ground motion defined in Equation (29) is discussed. The two components of the phase-shifted ground motion are

$$
\begin{cases}
a_{gX}(t, \Delta\phi_0) = \sum_{n=-N+1}^{N-1} c_{X,n} \exp[i\{\omega_n t - \mathrm{sgn}(\omega_n)\Delta\phi_0\}] \\
a_{gY}(t, \Delta\phi_0) = \sum_{n=-N+1}^{N-1} c_{Y,n} \exp[i\{\omega_n t - \mathrm{sgn}(\omega_n)\Delta\phi_0\}]
\end{cases}
. \tag{A30}
$$

The diagonal term of Equation (A26) for the phase-shifted ground motion is

$$
\begin{aligned}
&\int_0^{t_d} \{a_{gX}(t, \Delta\phi_0)\}\{a_{gY}(t, \Delta\phi_0)\} dt \\
&= t_d \sum_{n=-N+1}^{N-1} c_{X,n} c_{Y,-n} \exp\{-i\mathrm{sgn}(\omega_n)\Delta\phi_0\} \exp\{-i\mathrm{sgn}(\omega_{-n})\Delta\phi_0\} \\
&= t_d \sum_{n=-N+1}^{N-1} c_{X,n} c_{Y,-n} \\
&(\because \exp\{-i\mathrm{sgn}(\omega_n)\Delta\phi_0\} \exp\{-i\mathrm{sgn}(\omega_{-n})\Delta\phi_0\} = \exp(-i\Delta\phi_0)\exp(i\Delta\phi_0) = 1)
\end{aligned}
\tag{A31}
$$

Therefore, from Equation (A27), the non-diagonal term of Equation (A26) for the phase-shifted ground motion is

$$
\int_0^{t_d} \{a_{gX}(t, \Delta\phi_0)\}\{a_{gY}(t, \Delta\phi_0)\} dt = 0. \tag{A32}
$$

Equation (A32) indicates that the principal axis of the phase-shifted horizontal ground motions $\mathbf{a_g}(t, \Delta\phi_0)$ coincides with that of the original ground motion $\mathbf{a_g}(t)$.

### Appendix D. Details of the Original Ground Motion Records Used in the Present Sudy

Table A1 shows the date of the event, the magnitude (Meteorological Agency Magnitude $M_J$, or moment magnitude $M_W$), location of epicenter, distance and station name of each record.

**Table A1.** Event date, magnitude, location of epicenter, distance, and station name of each record.

|   | ID | Event Date | Magnitude | Distance | Station Name |
|---|-----|-----------|-----------|----------|--------------|
| a | JKB | 1995/01/17 | $M_J = 7.3$ | 16 km | Kobe JMA Observatory |
| b | MKW | 2018/09/06 | $M_J = 6.7$ | 14 km | K-Net Mukawa (HKD126) |
| c | MSK | 2016/04/16 | $M_J = 7.3$ | 12 km | KiK-Net Mashiki (KMMH16) |
| d | SYL | 1994/01/17 | $M_W = 6.7$ | 5.3 km * | Sylmar—Olive View Med FF |
| e | TCU | 1999/09/20 | $M_W = 7.6$ | 0.89 km * | TCU075 |
| f | YPT | 1999/08/17 | $M_W = 7.5$ | 4.83 km * | Yarimca |
| g | HAC | 1968/05/16 | $M_J = 7.9$ | 177 km | Hachinohe Harbor |
| h | TOM | 2003/09/26 | $M_J = 8.0$ | 225 km | K-Net Tomakomai (HKD129) |

* This distance is the closest distance from rupture plane defined in the Pacific Earthquake Engineering Research Center (PEER) database, while the others from the Japanese database are the epicenter distance.

Figure A1 shows the time-history of original ground motion records and normalized Arias intensity $I_A(t)$ used in the present study. The Arias intensity at time $t$ is calculated as

$$
I_A(t) = \frac{\pi}{2g} \int_0^t \left[ \{a_{gX}(\tau)\}^2 + \{a_{gY}(\tau)\}^2 \right] d\tau. \tag{A33}
$$

In this figure, the effective duration $t_{D5-95}$ is defined 5–95% of normalized $I_A(t)$ calculated from horizontal ground acceleration, following Trifunac and Brandy [69]. Note that some of the ground motions shown here may have non-zero acceleration records outside the range (e.g., (a) JKB after 80 s). Although the whole as-provided records without

any trimming are used in this study, the parts not shown in these figures have little contributions to the analysis results because they are close to zero.

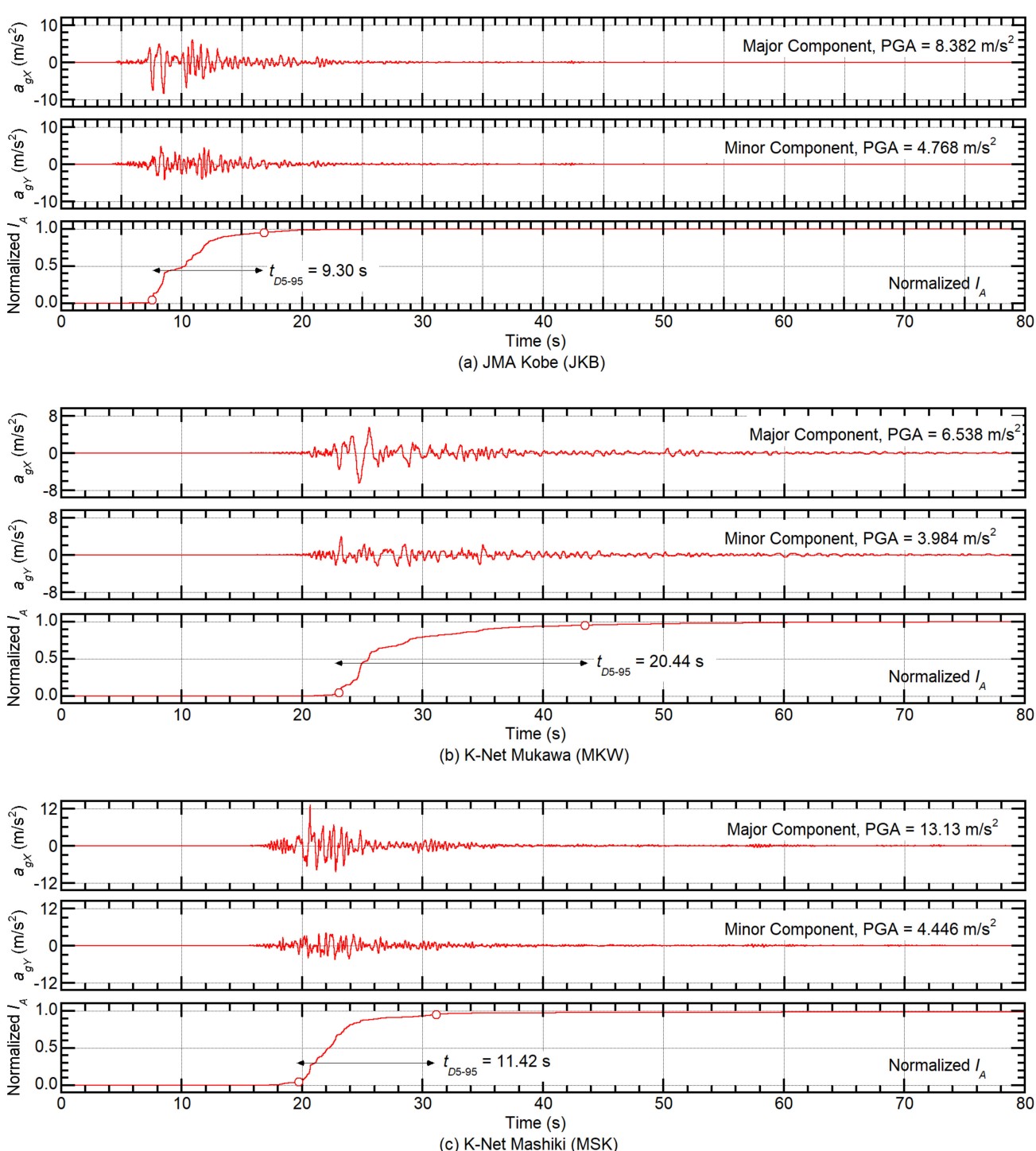

**Figure A1.** *Cont.*

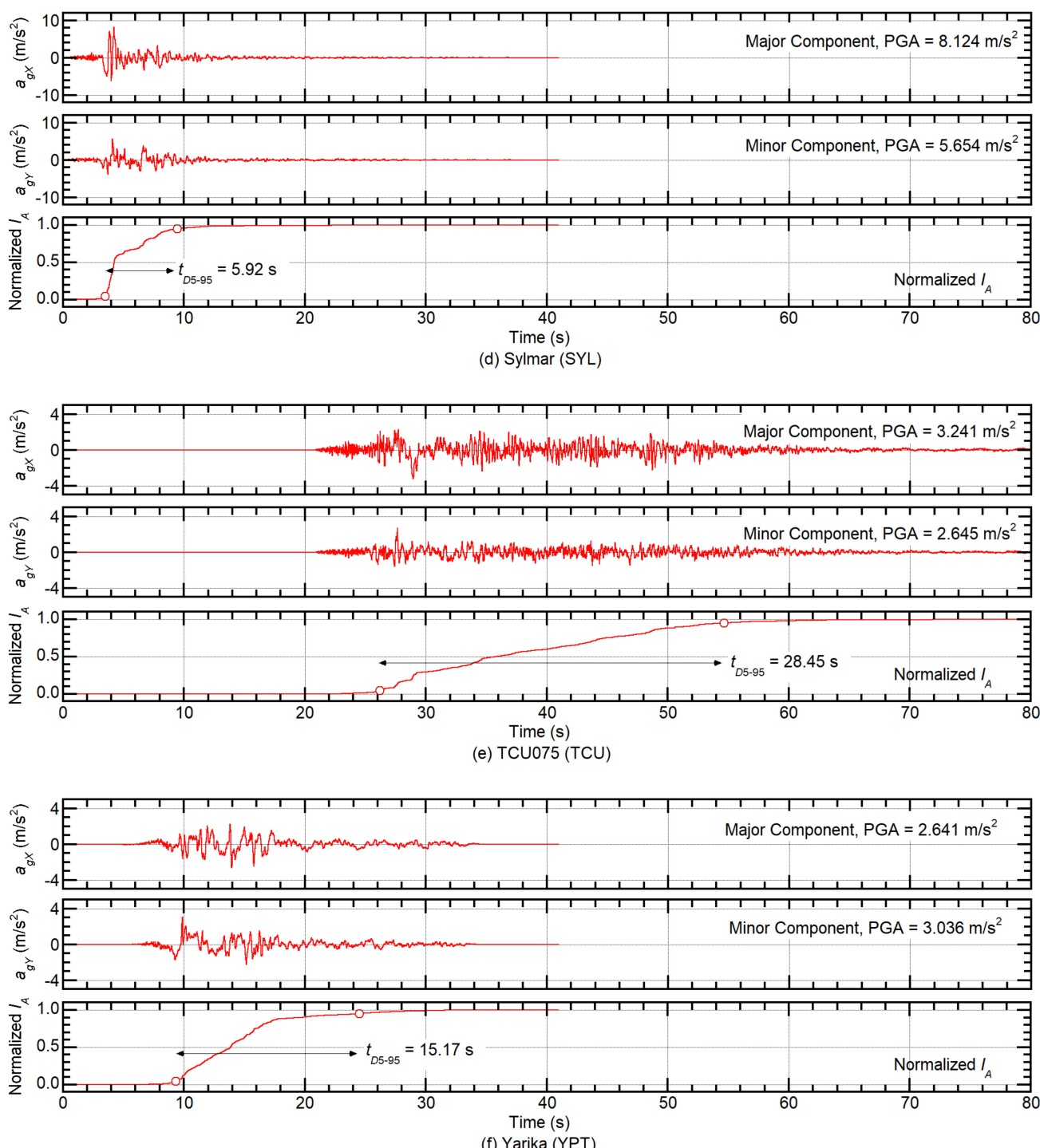

**Figure A1.** *Cont.*

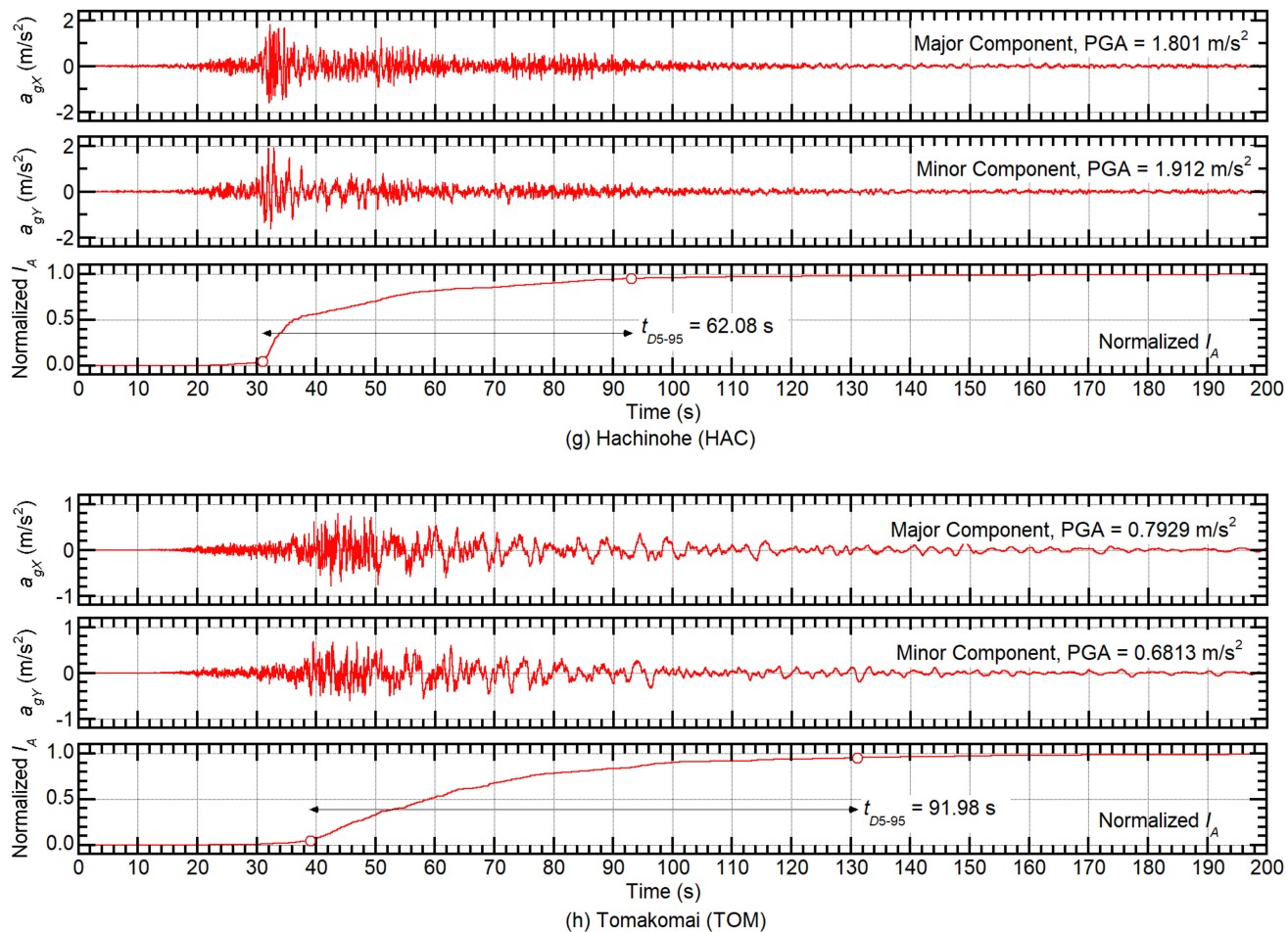

**Figure A1.** Ground motion time histories and normalized Arias intensity of original records.

## Appendix E. Comparisons of the Time-History of Phase-Shifted Ground Motions

In Appendix E, the time-history of phase-shifted ground motions are compared to see the local difference. Figure A2 shows the comparisons of the time-history of orbit of phase-shifted ground motions (JKB) as an example. As shown in Figure A2a,b, the time-histories of the X- and Y- components are locally different, although the whole waveforms are similar. In addition, their orbits are similar to the original ground motion ($\Delta\phi_0 = 0$), as shown in Figure A2c.

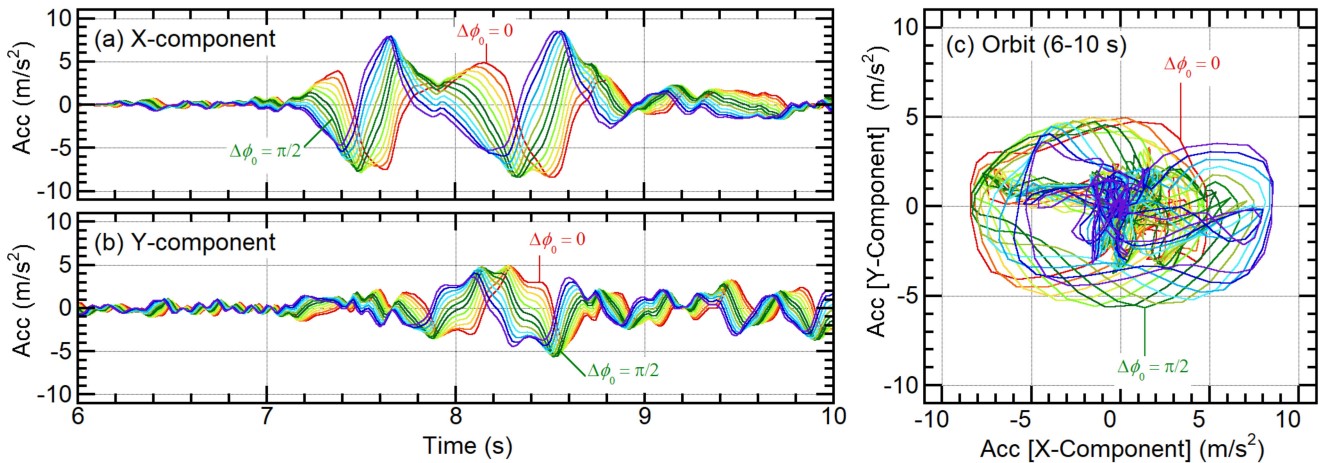

**Figure A2.** Comparisons of the time-history and orbit of phase-shifted ground motions (JKB).

## Appendix F. Formulation of Equation (31)

Assuming that the damping is proportional to the tangent stiffness and the initial damping ratio $h_0$ is 0.05, the cumulative dissipated energy during a half cycle can be expressed as:

$$\Delta E_\mu = Q_y d_y f(\mu) \approx (0.85)^2 \Delta E_{\max}. \tag{A34}$$

Therefore, $Q_y d_{\max} / \Delta E_{\max}$ can be approximated as:

$$\frac{Q_y d_{\max}}{\Delta E_{\max}} \approx \frac{Q_y d_y \mu}{Q_y d_y f(\mu) / (0.85)^2} = (0.85)^2 \frac{\mu}{f(\mu)}. \tag{A35}$$

Equation (A35) is identical to Equation (31).

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
