# Peer review of "Bidirectional Seismic Energy Input to an Isotropic Nonlinear One-Mass Two-Degree-of-Freedom System"

_buildings, doi:10.3390/buildings11040143_

Round 1

Reviewer 1 Report

Manuscript presents study about Bidirectional Seismic Energy Input to Isotropic Nonlinear One- mass Two-Degree-Of-Freedom System.

My comments are:

  • in Abstract Fajfar reference should be avoided, and "gamma" should be explained in first appearance not in line 43
  • chapter 2 is very good explained
  • in Table 1, what are accelerations of this motions if accelaration is used? Difference among data length and duration should be explained.
  • Disscuison chapter should be extended using introduction references - it is too general
  •  

Reviewer 2 Report

Please find attached document.

Round 2

Reviewer 2 Report

thank you for the replies/edits, please find report attached

Author Response

Please see the attachment/
